# Identification of a targetable *KRAS*-mutant epithelial population in non-small cell lung cancer

Giorgia Maroni[1,2,3,22], Mahmoud A. Bassal [1,2,22], Indira Krishnan[2,22], Chee Wai Fhu[1], Virginia Savova[4], Rapolas Zilionis[4,5], Valerie A. Maymi[6,7], Nicole Pandell[6,7], Eva Csizmadia[6], Junyan Zhang[2], Barbara Storti[8], Julio Castaño [9], Riccardo Panella[2,10], Jia Li[1], Corinne E. Gustafson [11], Sam Fox[11], Rachel D. Levy[11], Claire V. Meyerovitz[11], Peter J. Tramontozzi[11], Kimberly Vermilya[11], Assunta De Rienzo[2,11], Stefania Crucitta[12], Daniela S. Bassères[13], Marla Weetall[14], Art Branstrom[14], Alessandra Giorgetti[15,16], Raffaele Ciampi[17], Marzia Del Re[18], Romano Danesi [12], Ranieri Bizzarri[8,19], Henry Yang[1], Olivier Kocher[2,6], Allon M. Klein [4], Robert S. Welner[20], Raphael Bueno[2,11], Maria Cristina Magli[3], John G. Clohessy [2,6,7], Azhar Ali[1,23], Daniel G. Tenen [1,2,21,23✉] & Elena Levantini [2,3,6,21,23✉]

Lung cancer is the leading cause of cancer deaths. Tumor heterogeneity, which hampers development of targeted therapies, was herein deconvoluted via single cell RNA sequencing in aggressive human adenocarcinomas (carrying *Kras*-mutations) and comparable murine model. We identified a tumor-specific, *mutant-KRAS-associated* subpopulation which is conserved in both human and murine lung cancer. We previously reported a key role for the oncogene BMI-1 in adenocarcinomas. We therefore investigated the effects of in vivo PTC596 treatment, which affects BMI-1 activity, in our murine model. Post-treatment, MRI analysis showed decreased tumor size, while single cell transcriptomics concomitantly detected near complete ablation of the *mutant-KRAS-associated* subpopulation, signifying the presence of a pharmacologically targetable, tumor-associated subpopulation. Our findings therefore hold promise for the development of a targeted therapy for *KRAS-mutant* adenocarcinomas.

[1] Cancer Science Institute of Singapore, National University of Singapore, Singapore, Singapore. [2] Harvard Medical School, Boston, MA, USA. [3] Institute of Biomedical Technologies, National Research Council (CNR), Area della Ricerca di Pisa, Pisa, Italy. [4] Department of Systems Biology, Harvard Medical School, Boston, MA, USA. [5] Institute of Biotechnology, Life Sciences Center, Vilnius University, Vilnius, Lithuania. [6] Beth Israel Deaconess Medical Center, Boston, MA, USA. [7] Preclinical Murine Pharmacogenetics Core, Beth Israel Deaconess Cancer Center, Dana Farber/Harvard Cancer Center, Boston, MA, USA. [8] NEST, Scuola Normale Superiore and Istituto Nanoscienze-CNR, Pisa, Italy. [9] Platform for Immunotherapy BST-Hospital Clinic, Banc de Sang i Teixits (BST), Barcelona, Spain. [10] Center for Genomic Medicine, Desert Research Institute, Reno, NV, USA. [11] Division of Thoracic Surgery, The Lung Center and the International Mesothelioma Program, Brigham and Women's Hospital, Boston, MA, USA. [12] Unit of Clinical Pharmacology and Pharmacogenetics, Department of Clinical and Experimental Medicine, University of Pisa, Pisa, Italy. [13] Biochemistry Department, Chemistry Institute, University of Sao Paulo, Sao Paulo, Brazil. [14] PTC Therapeutics, 100 Corporate Court, South Plainfield, NJ, USA. [15] Cell Biology Unit, Department of Pathology and Experimental Therapeutics, Faculty of Medicine and Health Sciences, University of Barcelona, Barcelona, Spain. [16] Stem Cell Biology and Leukemiogenesis Group, Regenerative Medicine Program, Institut d'Investigació Biomèdica de Bellvitge - IDIBELL, L'Hospitalet de Llobregat, Barcelona, Spain. [17] Endocrine Unit, Department of Clinical and Experimental Medicine, University Hospital of Pisa, Pisa, Italy. [18] Unit of Clinical Pharmacology and Pharmacogenetics, Department of Laboratory Medicine, University Hospital of Pisa, Pisa, Italy. [19] Department of Surgical, Medical and Molecular Pathology, and Critical Care Medicine, University of Pisa, Pisa, Italy. [20] University of Alabama at Birmingham, Department of Medicine, Hemathology/Oncology, Birmingham, AL, USA. [21] Harvard Stem Cell Institute, Cambridge, MA, USA. [22] These authors contributed equally: Giorgia Maroni, Mahmoud A. Bassal, Indira Krishnan. [23] These authors jointly supervised this work: Azhar Ali, Daniel G. Tenen, Elena Levantini. ✉email: daniel.tenen@nus.edu.sg; elevanti@bidmc.harvard.edu

N on-small cell lung cancer (NSCLC), the most common epithelial tumor, comprising ~85% of pulmonary malignancies, is the leading cause of cancer-related deaths[1]. Considerable heterogeneity exists among lung adenocarcinomas (ADCs). Among the genes implicated in their etiology[2], frequent activating mutations in KRAS have been identified in 10–30% of cases. In addition, loss-of-function mutations in p53 occur in ~50–70% of cases[3] and co-occur with KRAS mutations in ~40% of cases[4]. Besides direct covalent KRAS-G12C inhibition[5], no therapies have been approved for mutant-KRAS NSCLCs[4]; therefore identification of tumorigenic subpopulations sustaining growth may contribute to improved targeted therapies.

Resolving the distinct subpopulations of healthy versus tumor-bearing lungs has been hampered by traditional ensemble-based methods such as bulk RNA sequencing, and gaps-in-knowledge on specific phenotypic markers. Recently, single-cell RNAseq (sc-RNAseq) has enabled analysis of complex tissues and characterization of cellular identity, by grouping cells based on their gene expression profiles, at an unprecedented high-resolution[6].

Pulmonary sc-RNAseq on tumor epithelial cells represents an undeveloped field. A pioneering study on fluorescence-activated cell sorting-purified murine lungs distinguished healthy multipotential, bipotential, and mature alveolar type II (ATII) epithelial cells[7]. Subsequently, identification of markers for major normal body-wide lineages gave rise to the mouse cell atlas (MCA)[8] with similar efforts currently underway for humans as part of the Human Cell Atlas[9–11]. Pulmonary-associated immune cells in healthy[12], inflamed[13], or transformed lungs[14–16] have been identified in both human and murine tissues, including our study comparing tumor-infiltrating myeloid subpopulations in both species NSCLCs[17].

Although tumor heterogeneity hampers major therapeutic advancements, little is known on how transformation events orchestrate molecular/cellular alterations within lung cancer. Our deconvolution of human NSCLCs leads to the identification of a distinct epithelial subpopulation, selectively detectable in ADCs carrying the aggressive mutant-KRAS oncogene.

We also comprehensively mapped pulmonary subpopulations in normal and tumor-bearing lungs, by adopting a model of ADC (Kras$^{+/G12D}$;Trp53$^{-/-}$, henceforth referred to as KP), which combines Kras activation with p53 ablation in pulmonary epithelium[18–20]. Our data produced a unique cellular atlas of healthy lungs and KP ADCs, and found new cell subtypes that are distinctly associated with disease. Newly identified tumor-enriched subpopulations were discovered, of which one represents a novel specific epithelial tumor cluster, matching a signature of markers that we also selectively identified in the human mutant-KRAS-specific subpopulation. Both murine and human mutant-KRAS-specific subpopulations are positive for the oncogene Bmi-1 (B-cell-specific Moloney murine leukemia virus integration site 1), a key component of the epigenetic complex polycomb repressive complex-1, which belongs to the 11-gene death-from-cancer-signature[21]. Since its discovery, BMI-1 has been implicated in several biological phenomena including development, cell cycle, DNA damage response, senescence, stem cell, self-renewal, and cancer. BMI-1 has recently proven to be of significant clinical relevance as it overexpressed in a number of malignancies[22–30]. We previously identified BMI-1 as a critical druggable target in NSCLC[31]. Here, we tested on KP mice PTC596, a drug identified by its ability to eliminate BMI-1$^+$ leukemic cells[32] and currently in phase (Ph) 1b trial (Identifier NCT02404480) for solid malignancies. As assessed by magnetic resonance imaging (MRI), PTC596 treatment demonstrated more rapid and efficient antitumor ability than conventional therapy. sc-RNAseq, depicting the transcriptional dynamics encompassing tumor response to PTC596, emphasized a strong decrease of the epithelial subpopulations as well as the tumor-specific epithelial cluster, suggesting Kras-mutant tumor is amenable to PTC596 treatment. PTC596 is also capable of decreasing tumor growth of human mutant-KRAS xenograft models, encouraging the development of PTC596-based therapies for NSCLC patients carrying KRAS mutations for which no pharmacological indication is available.

## Results

### sc-RNAseq deconvolution of human NSCLCs unravels tumor heterogeneity between wild-type and mutant KRAS ADCs.
To study the epithelial component constituting human NSCLCs, we performed sc-RNAseq analysis on freshly isolated biopsies[17] from 12 patients (Supplementary Table 1). Once inter-sample and batch variability was accounted for, defined subpopulations were identified using SingleR[33], which used the annotated Human Primary Cell Atlas[11] data set for reference cell signatures. Despite the typical high genomic variability of human NSCLCs, we were able to identify common subpopulations, which we visualized utilizing SPRING[34]. The force-directed layout of k-nearest-neighbor graphs depicted 15 distinct transcriptional clusters (C1–C15) (Fig. 1a). Specifically, we identified the epithelial compartment (Fig. 1b), and distinct subpopulations were also identified, which contain tumor-infiltrating immune cells (Fig. 1c), endothelial cells (Fig. 1d), and fibroblasts (Fig. 1e).

By splitting only the ADC samples according to their KRAS mutation status (mutants n = 8, wt = 2), we discovered the presence of a transcriptional epithelial cluster, C10, which was almost exclusively detected in mutant-KRAS ADCs alone (false discovery rate (FDR) = $2.07 \times 10^{-116}$) (Fig. 1f, Supplementary Figures 1a–c, Supplementary Table 2, and Supplementary Data), urging us to ask whether it may represent a novel subpopulation capable of distinguishing the most aggressive and still untargetable mutant-KRAS ADCs from the KRAS WT ADCs.

### sc-RNAseq of healthy and tumor-bearing lungs highlights the presence of a transformed epithelial tumor-specific cluster.
Having identified a specific epithelial tumor subpopulation in human mutant-KRAS ADCs, this prompted us to ask whether we may detect its murine counterpart, and subsequently attempt its in vivo targeting, by employing KP mice that develop pulmonary ADCs[20].

By performing sc-RNAseq on normal lungs from control littermates and KP tumors, we identified which subpopulations were present in tumors versus healthy lungs. Algorithmically, 13 clusters (C1–C13) were identified (Fig. 2a). These clusters were transcriptionally distinct, as shown by the top 25 marker genes per cluster in both healthy (heatmaps and gene lists in Supplementary Figure 2a and Supplementary Table 3, respectively) and tumor tissues (Supplementary Figure 2b and Supplementary Table 4). By comparing lung tumors with healthy lungs by cell number quantification, we found that C3, C4, C7, and C11 were significantly underrepresented in tumors (FDR C3 = 0, C4 = $4.42 \times 10^{-10}$, C7 = $1.47 \times 10^{-288}$, C11 = $1.12 \times 10^{-96}$), whereas clusters C1, C2, C5, C8, C10, C12, and C13 were enriched (FDR C1 = $2.46 \times 10^{-3}$, C2 = $4.22 \times 10^{-56}$, C5 = $4.39 \times 10^{-15}$, C8 = $4.92 \times 10^{-95}$, C10 = $1.36 \times 10^{-192}$, C12 = $1.05 \times 10^{-72}$, C13 = $1.22 \times 10^{-289}$), as compared with control lungs (Fig. 2b and Supplementary Table 5). C9 instead was equally distributed in both tissues (FDR $6.58 \times 10^{-1}$). Clusters C2, C10, and C13 were almost virtually exclusive to tumors, being barely detectable in healthy lungs, and therefore representing epithelial tumor-enriched clusters (TECs) (Figs. 2a and 2b, Supplementary Figure 2c).

In addition, to classify the cell types within each cluster, a hybrid annotation strategy was employed wherein immunological

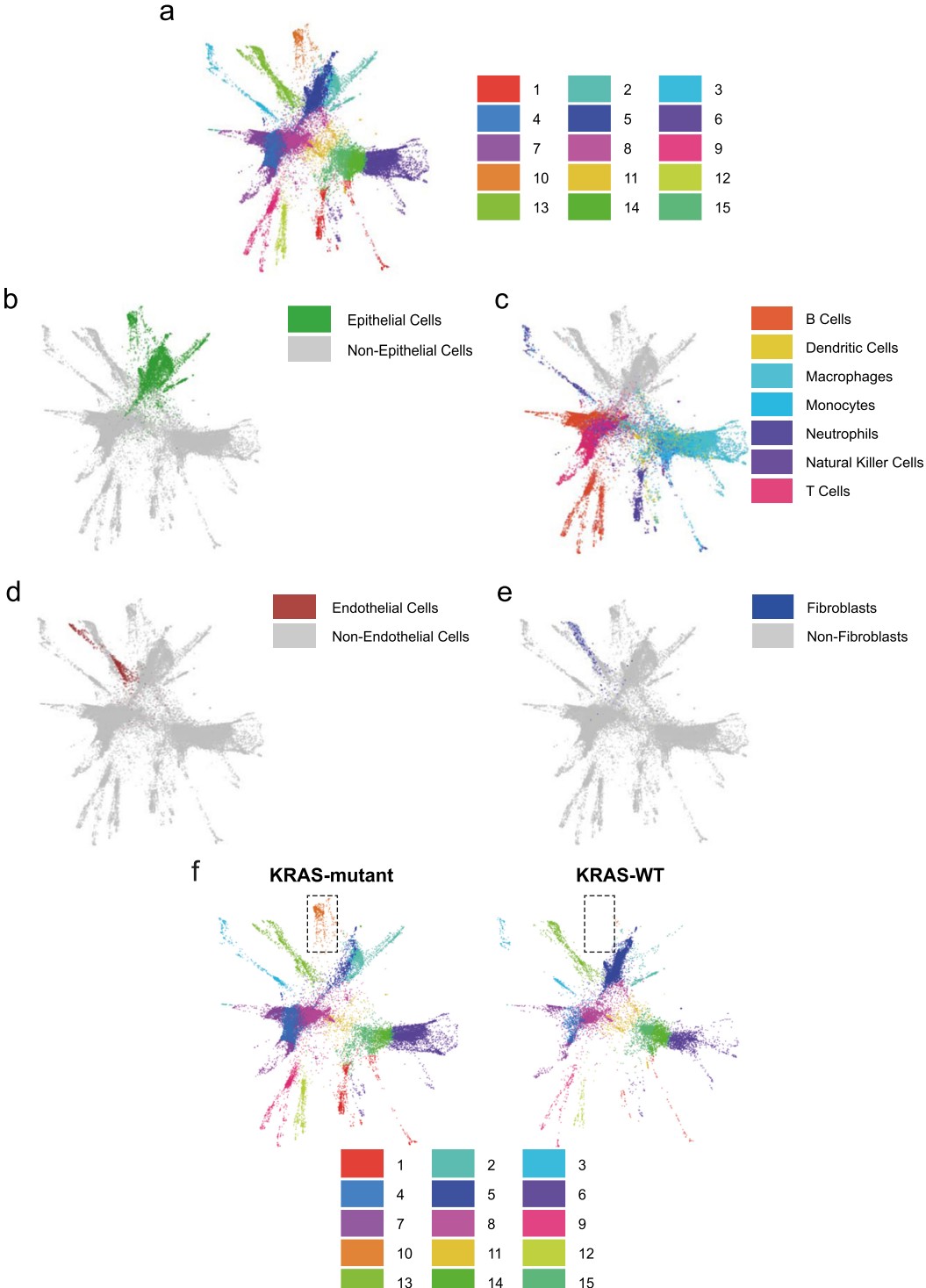

**Fig. 1 Single-cell RNA sequencing deconvolution of human NSCLC unravels tumor heterogeneity between WT and mutant *KRAS* adenocarcinomas.**
**a** SPRING plot of the 15 human clusters identified in 12 NSCLCs. Each point represents one cell. Each color represents a defined transcriptomic cluster.
**b** Epithelial cells (green); **c** immune cells (panel legend defines subpopulations); **d** endothelial cells (maroon), and **e** fibroblasts (blue) were identified.
**f** SPRING plots of the 15 clusters in ten ADC patients (eight carrying *KRAS* mutations, two wildtypes for *KRAS*). Dotted squares highlight cluster 10 (C10) is unique to the *KRAS*-mutant samples.

cells were identified using the Immgen data set[35] via SingleR, and the remaining cells annotated using MCA transcriptional maps[8] to leverage the strengths of each annotation data set. This approach enabled accurate detection of numerous epithelial cell types (Fig. 2c), namely alveolar bipotent progenitors, alveolar type I (ATI), ATII, basal epithelial, ciliated, club as well as endothelial,

immune, and fibroblast subpopulations (Fig. 2d). Cell number quantification revealed that ATII-like cells are the most enriched within the tumor milieu expanding to 31.7%, as compared with 1.2% in healthy tissues (Supplementary Table 6).

By superimposing cell annotations (Fig. 2d) on the cluster distribution (Fig. 2a), our analysis revealed an unprecedented

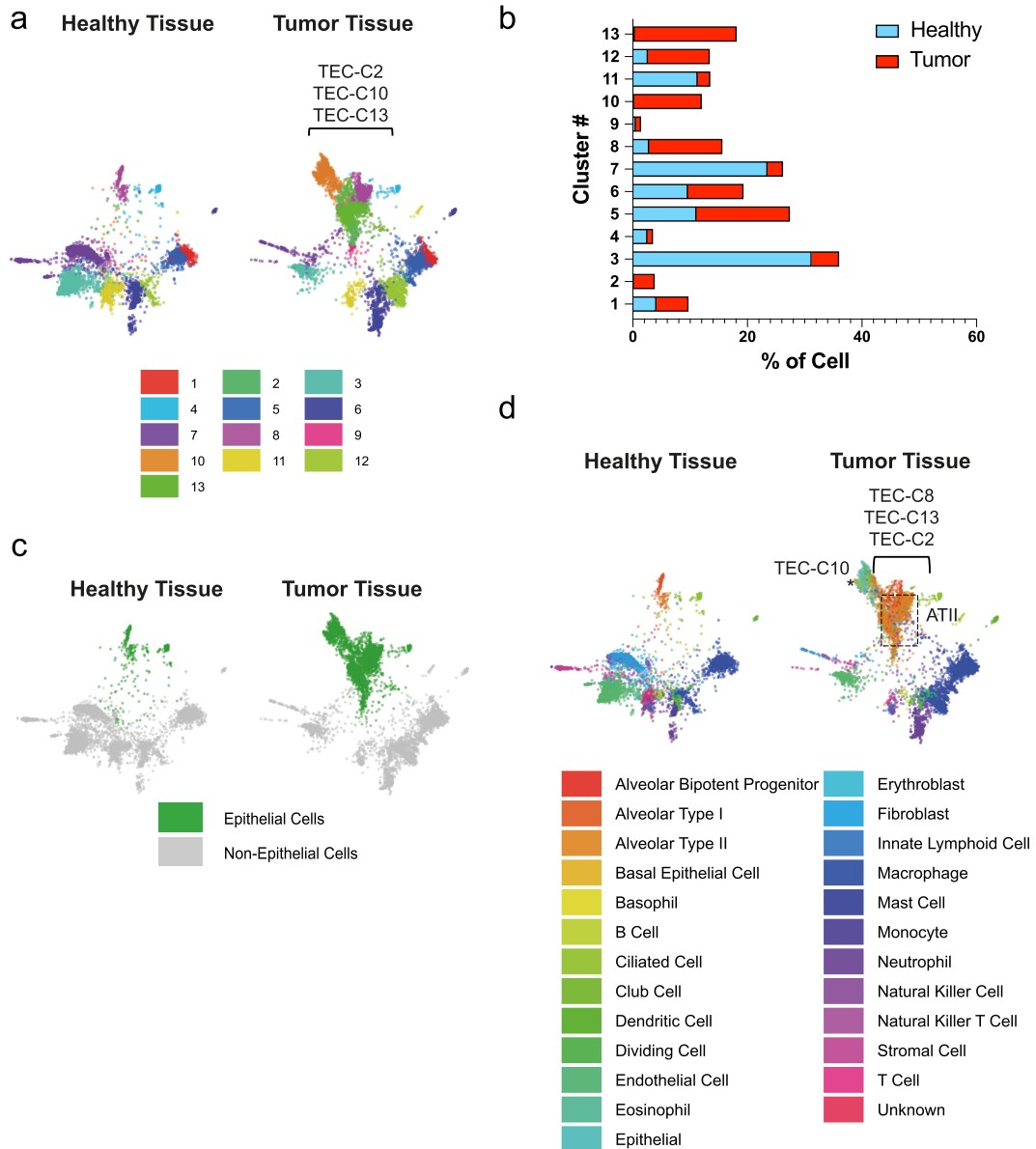

**Fig. 2 Single-cell RNA sequencing of healthy and tumor-bearing lungs highlights the presence of a transformed epithelial tumor-specific cluster.**
**a** SPRING plots of the 13 clusters identified in healthy (*n* = 2, left panel) and tumor tissues (*n* = 3, right panel). **b** Percentages of clusters distributions in healthy (blue) and tumor (red) tissues. **c** SPRING plots showing the epithelial compartment (green) in healthy and tumor tissues. **d** SPRING plots showing the annotated cell types (panel legend) in healthy and tumor tissues. * labels TEC-C10; the dotted boxes highlight ATII-like cells.

definition of the healthy and tumor epithelial compartment, wherein we observed that healthy ATII-like cells are comprised of two distinct transcriptomic profiles, defined as C13 (0.3%) and the slightly more abundant C8 (2.8%) (Fig. 2d, and Supplementary Table 5). Interestingly, in the tumor, in addition to the enrichment of these two ATII-like clusters (TEC-C13 = 17.8% and TEC-C8 = 12.8%), an additional cluster (TEC-C2) can also be annotated as ATII-like. Furthermore, TEC-C10, while being annotated as positive for a generic "epithelial" cell signature (Fig. 2d, right panel and labeled by the *), did not match any normal pulmonary epithelial subpopulation, therefore representing a bona fide transformed TEC/cell subpopulation. Overall, our data identified KP tumor-enriched epithelial cells, which are mainly comprised of the tumor-specific subpopulation (C10), which does not match any normal epithelial signature, as well as ATII cells, which themselves consist of different clusters, as described above.

Having identified C10 as being virtually unique to tumor tissues and not ascribable to any common, defined epithelial cell type, we attempted to further characterize its differences over the other tumor epithelial clusters. Transcriptionally, the top 50 highest and lowest expressing genes are sufficient to show the unique transcriptional profile of C10, as compared with all other tumor epithelial clusters (Fig. 3a). The genes represented in the heatmap are listed in Supplementary Table 7. Gene expression profiling and gene set enrichment analysis (GSEA) of C10 cells versus all other tumor epithelial cells, showed enrichment of (i) stem cell signatures (Fig. 3b), (ii) stemness genes (stem cells, embryonic, mammary stem cells, liver cancer stem cells), cancer radiotherapy responsiveness (Supplementary Figure 2d), as well as target genes of the cancer stem cell gene BMI-1 (Fig. 3b and Supplementary Figure 2d). Consistently, RNA velocity analysis, which attempts to elucidate cell's transcriptomic differentiation trajectory or "direction"[36], showed that C10 contains a

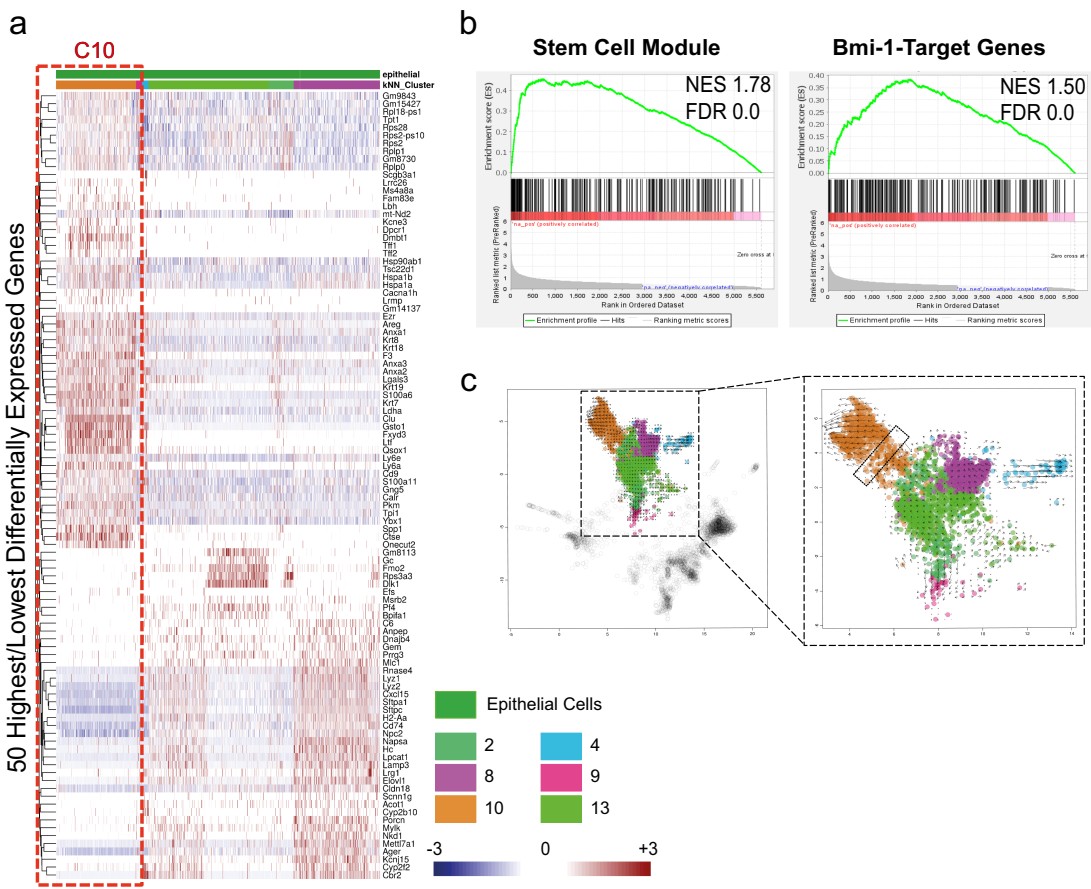

**Fig. 3 The transformed epithelial tumor-specific cluster shows a defined signature. a** Heatmap showing the 50 highest and lowest expressed genes in C10. **b** GSEA shows enrichment of stem cell module genes (FDR = 0.0) and Bmi-1 target genes (FDR = 0.0) in C10, as compared with all other tumor epithelial clusters. Normalized enrichment score (NES) is indicated in each panel. **c** RNA velocity analysis recapitulates dynamics of tumor epithelial cells (zoomed in the dotted box) differentiation. Within the dotted box short/no arrow-containing cells indicate the point of origin from which other C10 (orange) tumor cells are originated.

transcriptomic point of origin (short/no arrows; Fig. 3c) from which, the other tumor epithelial cells contained within C10 stem. These data, therefore, support the hypothesis that C10 potentially contains tumor-initiating cells.

**Human and murine KP ADCs display a tumor cluster specifically associated with harboring KRAS mutations.** After comparing differentially expressed genes between mutant-*KRAS*-associated C10 clusters, in both humans and mice, versus all other epithelial clusters, we identified a common mutant-KRAS-associated signature of 21 upregulated (LogFC ≥ 1.5) and nine downregulated (LogFC ≤ −1.5) homologous markers (Fig. 4a), which is specific for C10. SPRING plots display the colocalization of the combined upregulated (Fig. 4b, Supplementary 3a) and downregulated (Fig. 4c, Supplementary 3a) markers comprising the signature, thereby showing conservation across species. Representative expressions of selected marker genes contained within the signature are depicted in Figs. 4d and 4e. In particular, they represent genes contained within the 50 highest/upregulated (Fig. 4d) or lowest/downregulated (Fig. 4e) genes, commonly present in both the human and murine data sets. Upregulated genes contained within the conserved top 50 genes whose expression is highly restricted to C10 (Fig. 4d), act as oncogenes associated with poor prognosis in multiple human malignancies (Human Protein Atlas, www.proteinatlas.org[37] and[38–41]), supporting the aggressive nature of C10. In addition, among the conserved downregulated 50 genes C10 is more negatively labeled

by *SFTPC* and *HOPX* (Fig. 4e), the major markers for ATII and ATI cells, respectively, as well as other relevant differentiation markers of secretory (*SCGB3A1*) and ATII (*SFTPA1, NAPSA, SLC34A2, LPCAT1, LAMP3*) epithelial cells[42,43] (Fig. 4e). Overall, these data indicate that C10 represents a transformed subpopulation that does not match normal alveolar subtypes, thus representing a novel subpopulation capable of distinguishing mutant-*KRAS* from WT-*KRAS* ADCs. Of note, Ingenuity Pathway Analysis (IPA) identified EIF2, mTOR, eIF4/p706SK, and integrin signalings as the common enriched pathways within the top five ("cellular growth, proliferation, and development" category) in both murine and human C10 (Fig. 4f), which all act, according to the curated IPA software, downstream of activated *Kras* (Supplementary Figures 3b–e).

In addition, GSEA showed that murine C10-upregulated genes were significantly enriched within three different murine-curated data sets derived from samples displaying Kras upregulation (Supplementary Figure 3f). A similar result is shown by comparing human C10-upregulated genes to two curated data sets displaying upregulated KRAS signaling as well (Supplementary Figure 3g). Conversely, a comparison between the human C10 with both a data set containing genes overexpressed in NSCLCs genetically defined by copy number amplification and a Reactome data set of EGFR signaling (another main lung cancer driver) did not show any significant enrichment (Supplementary Figure 3h). Overall, these data confirm C10 is specifically associated with Kras signaling.

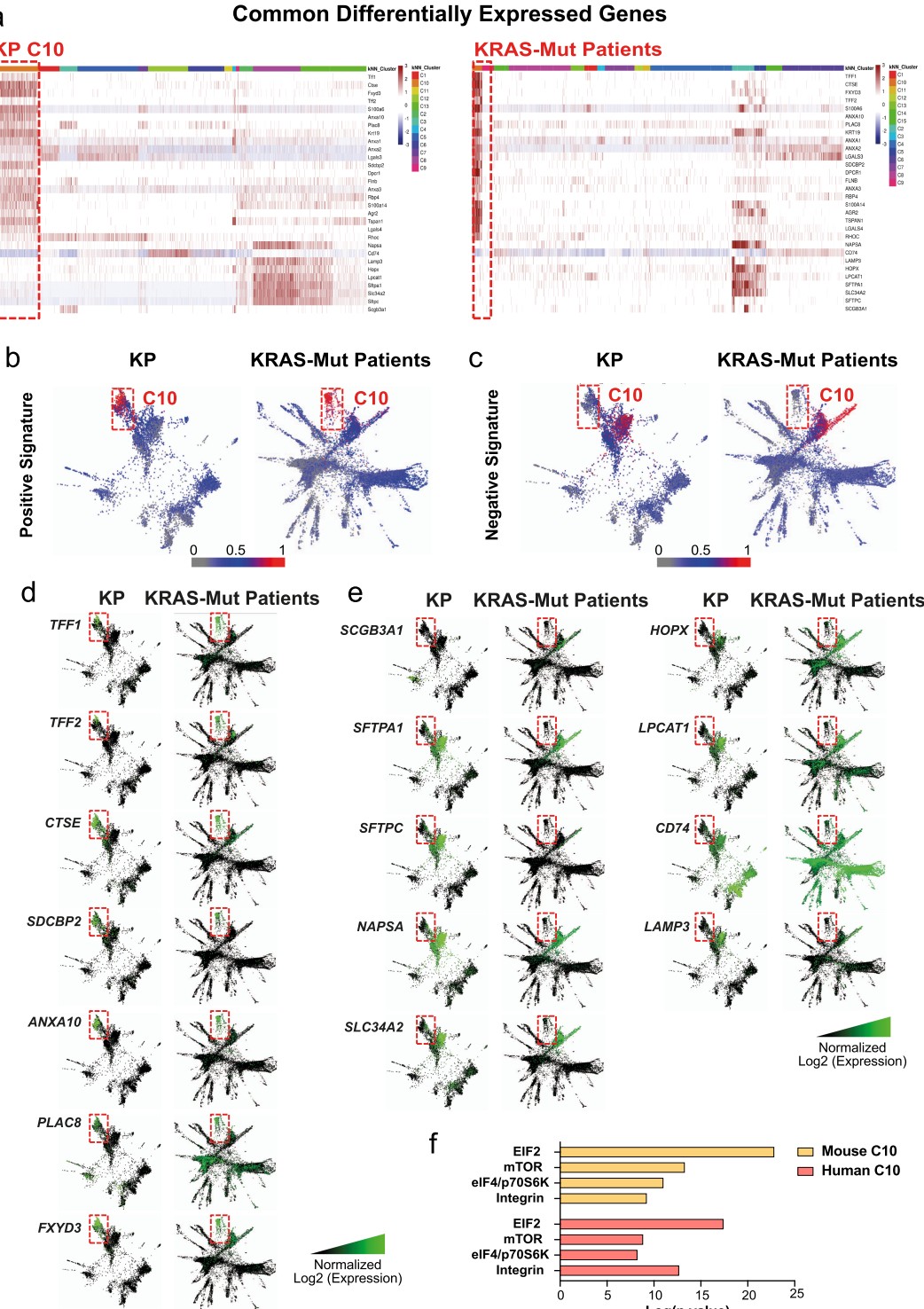

**Fig. 4 Human and murine KP ADCs display a tumor cluster specifically associated with harboring *KRAS* mutations. a** Heatmap of the common differentially expressed genes in both murine (upper panel) and human (lower panel) *KRAS*-mutated ADCs, as defined by differential gene expression analysis. Row-scaled *z* scores of the log2-normalized expression values per gene are shown. **b**–**c** SPRING plots showing the common signature enrichment score for C10 clusters, calculated for each cell equivalent to the number of detected genes from the common signature shown in **a**. For the positive/ upregulated (4**b**) and the negative/downregulated signature (4**c**) the more genes detected per cell, the stronger the enrichment score, represented as a scale from 0 (gray) to 0.5 (blue) to 1 (red), where an enrichment score of 1 signifies detected expression of all marker genes within that cell. SPRING plots showing a visual representation of the log2-normalized gene expression for the selected **d**, upregulated, and **e** downregulated genes of interest in both murine (left panels) and human (right panels) C10 clusters. **f** IPA analysis showing common top three enriched pathways in KP and Human C10s.

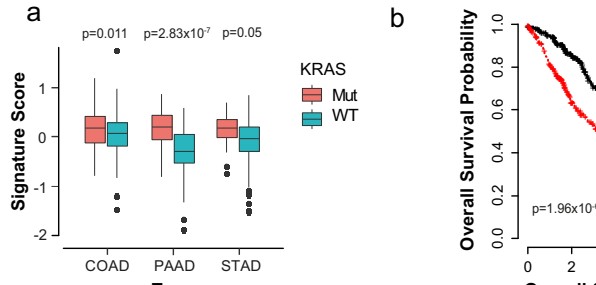

**Fig. 5 C10-specific signature is conserved in multiple tissue ADCs and it is associated with lower overall survival. a** Boxplots for distributions of the signature scores for combining the positive and negative signature genes over *KRAS*-mutant samples versus WT samples cross four adenocarcinomas (colon, COAD, $N = 100$ mutant *KRAS* and $N = 211$ WT; pancreas, PAAD, $N = 127$ mutant *KRAS* and $N = 31$ WT; stomach, STAD, $N = 18$ mutant *KRAS* and $N = 294$ WT). P values are indicated. **b** Survival analysis between C10 signature high and low groups ($p$ value $= 1.96 \times 10^{-6}$, hazard ratio $= 2.37$). C10 low group: $n = 223$, and C10 high group: $n = 222$.

In additionally, by utilizing the EnrichR enrichment suite, in both species, the molecular pathway downstream of the nuclear factor-κB subunit RelA/p65, which is required for *Kras*-induced lung tumorigenesis[19] are found to be enriched in C10, as compared with the other epithelial clusters (FDR = 0.01 and 0.03, in mouse and human data set, respectively) (Supplementary Table 8). *LGALS3, S100A6, AGR2*, and *TFF1*, which are contained within the mutant-KRAS-associated signature, are among the common RelA/p65-downstream effectors. IPA software also detected that both murine and human C10 display active Integrin, epidermal growth factor, insulin-like growth factor-1, extracellular signal-regulated kinase/mitogen-activated protein kinase, and insulin receptor pathways (Supplementary Figure 3i), as similarly found through proteogenomic network analysis on Kras[G12D] tumors (colon and pancreas)[44]. Consistently, IPA revealed that C10 in both species are enriched for signaling pathways involved in multiple malignancies associated with *KRAS* mutation, i.e., NSCLC, as well as colorectal cancer, pancreatic ADC, ovarian cancer, acute myeloid leukemia, melanoma, and endometrial cancer (Supplementary Figure 3j). Similarly, TCGA (The Cancer Genome Atlas) RNASeq samples belonging to several human ADCs [colon (COAD), pancreatic (PAAD), and stomach (STAD)], subdivided into mutant- and WT-*KRAS* status, show that the mutant-KRAS-associated signature identified is also applicable across cancer contexts ($p = 0.011$ for COAD, $N = 100$ mutant *KRAS* and $N = 211$ WT; $p = 2.83 \times 10^{-7}$ for the PAAD, $N = 127$ mutant *KRAS* and $N = 31$ WT; and $p = 0.05$ for STAD, $N = 18$ mutant *KRAS* and $N = 294$ WT) (Fig. 5a). Moreover, Kaplan–Meier survival analysis showed that overall, patients displaying high C10 signature had significantly poorer outcomes as compared with the group with low C10 signature ($p = 1.96 \times 10^{-6}$) (Fig. 5b), in line with the known aggressiveness of mutant KRAS tumors[45].

We have previously demonstrated, in a model of pulmonary ADC driven by CEBPα knockout in lung epithelial cells, the importance of Bmi-1[31], a major oncogene in NSCLC[30], with suggested roles in regulating cancer cells[31,46], and noteworthy, pharmacologically targetable[31]. Here, BMI-1 was positive by immunohistochemistry (IHC) in both murine and human lung tumors (Supplementary Figures 3k, 3m, and 3o). Most importantly, both murine and human C10 clusters, in which the mutant-KRAS-associated signature was identified, are positive for BMI-1 transcripts (Supplementary Figures 3l and 3n).

**PTC596 treatment of the mutant-KRAS A549 and SKLU1 cell lines affects their cell cycle progression.** Having discovered that *BMI-1* is expressed in both human and murine mutant-*KRAS*-associated clusters C10, we initially investigated the efficiency of

its pharmacologically driven downregulation in mutant-*KRAS* ADC cells. By collaborating with PTC Therapeutics, we utilized two of their compounds, PTC596 (Supplementary Figure 4a) and its analog PTC028 (Supplementary Figure 4b). PTC596, identified by its ability to kill BMI-1[+] cancer cells[32], is currently being tested in clinical trials for solid malignancies, as an orally bioavailable drug, which displays a long-circulating half-life, and lacks the multidrug transporter P-glycoprotein substrate activity[47]. PTC596 and PTC028 reportedly result in hyperphosphorylated BMI-1[32,48] associated with cell cycle arrest in $G_2$-M[47]. By treating A549 and SKLU1 cells with PTC596 (1 μM), PTC028 (1 μM) or vehicle control (0.5% dimethyl sulfoxide (DMSO)) for 24 h, 48 h, and 72 h, we confirmed by western blot presence of a band higher than 40–42 kDa at 24 h of drug treatment (Fig. 6a, b, Supplementary Figures 4c, d), corresponding to the hyperphosphorylated BMI-1 form[32,48]. Concurrent cell cycle analysis carried out at 24 h ($n = 3$), when the major effect on BMI-1 hyper-phosphorylation is observed, concomitantly reveals a significantly higher number of cells in $G_2$-M upon PTC596 (80.5% ± 3.7 $p = 4.52 \times 10^{-4}$ in A549; 69.9% ±2.8 $p = 4.58 \times 10^{-4}$ in SKLU1) and PTC028 (80.2% ±1.9 $p = 6.05 \times 10^{-6}$ in A549; 74.9% ± 3.1 $p = 6.02 \times 10^{-4}$ in SKLU1) treatment, as compared with DMSO-treated cells (15.2% ±1.2 in A549; 35% ±1.3 in SKLU1) (Fig. 6c–f).

BMI-1 acts as a major component of the chromatin remodeling complex PRC1, which ubiquitinates Histone H2A at lysine 119 (H2AK119ub). To determine whether PTC596 treatment affected BMI-1 activity, immunofluorescence staining for H2AK119ub, as readout for BMI-1 activity[49] was performed. A549 cells treated with the clinical-grade compound PTC596, which was adopted in the subsequent murine in vivo studies, show almost undetectable H2AK119ub nuclear staining, as compared with vehicle-treated cells ($n = 32$ each, mean±standard error fluorescence for PTC596-treated cells 170 ± 26 and vehicle-treated 2800 ± 110; $p = 2.90 \times 10^{-24}$), validating our pharmacological approach (Figs. 6g and 6h).

**PTC596 in vivo treatment affects the growth of KRAS-mutant ADCs.** As PTC596 and PTC208 treatments affect A549 cell cycle progression in culture, we tested their efficacy in in vivo tumor growth of human mutant-*KRAS* cells, by generating xenograft models of A549 cells in immunocompromised NSG mice. After subcutaneously injecting $2 \times 10^6$ cells per flank, once subcutaneous tumors reached ~80–90 mm³, we started bi-weekly treatment with PTC596 ($n = 15$), PTC028 ($n = 7$), or vehicle ($n = 20$) for 3 weeks. Figure 7a depicts the size of individually treated tumors at treatment termination, normalized to day 0. PTC596- and PTC028-treated xenografts showed a significant

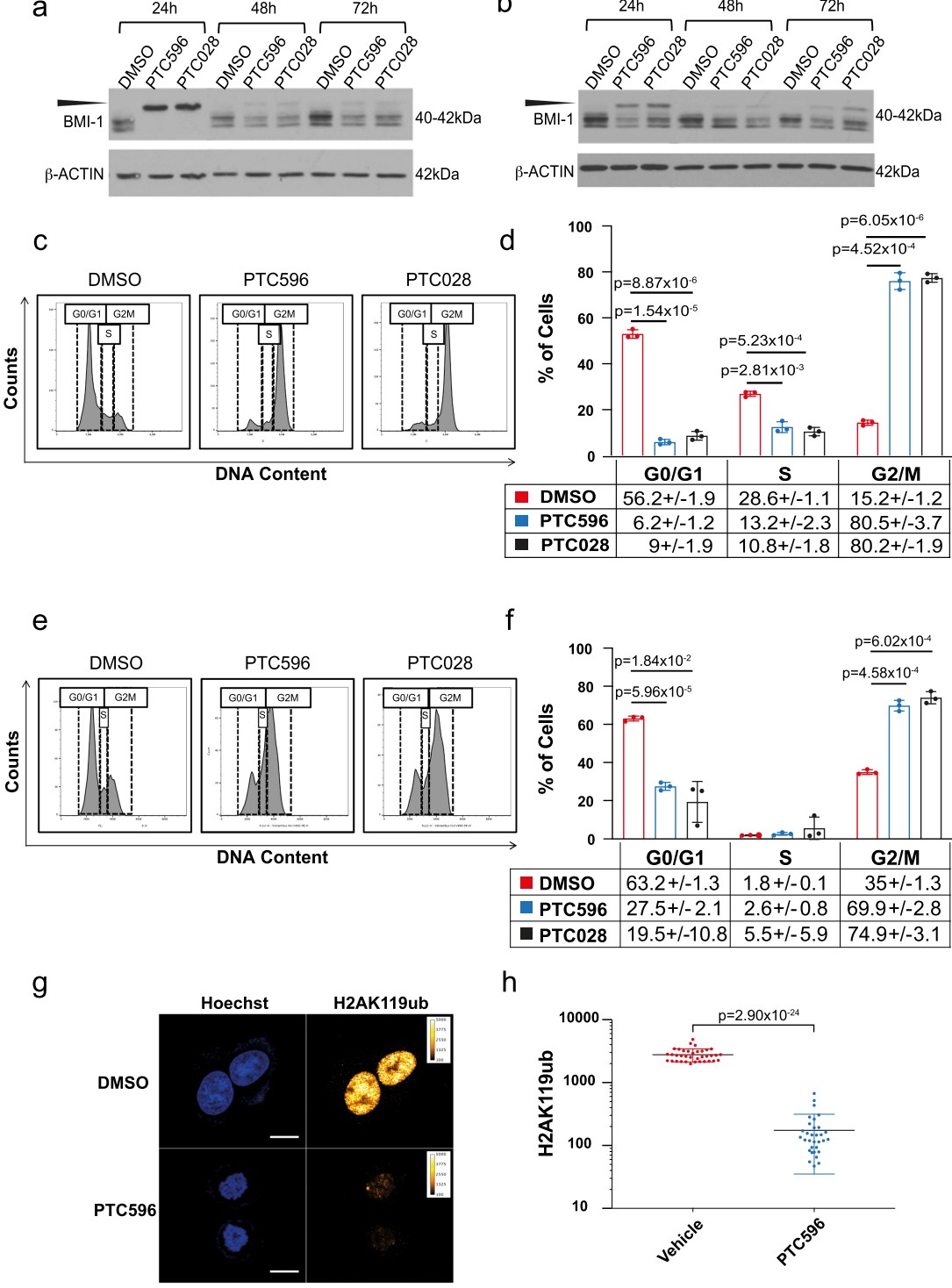

**Fig. 6 PTC596 treatment of the *KRAS* mutant A549 and SKLU1 cell lines affects their cell cycle progression.** Western blot analyses of human **a** A549 and **b** SKLU1 cell lines treated for 24, 48, and 72 hours with DMSO vehicle as control, and PTC596 or PTC028. Protein lysates were immunoblotted with an anti-BMI-1 antibody. Loading was assessed with an anti-β-actin antibody. The expected size is shown in kDa. The slower migrating hyperphosphorylated BMI-1 band is indicated by the arrowhead. **c–f** Cell cycle analysis of the A549 **c–d** and SKLU1 **e–f** cell lines after treatment for 24 hours with DMSO (red), PTC596 (blue), or PTC028 (black). The bar charts (**d** and **f**) represent the distribution of cells in $G_0$-$G_1$, S, and $G_2$-M phases. P values are indicated. Error bars represent standard deviation (SD). Percentages of cells in each cell cycle phase are indicated. **g** Fluorescence nuclear imaging of DNA and H2AK119ub in A549 cell line by confocal microscopy. Upper panels show DMSO-treated cells; lower panels show cells treated with PTC596 for 24 hours. Left panels show DNA staining by Hoechst 33342 (blue acquisition chanel). Right panels show H2AK119ub staining by immunofluorescence (scale bar 10 μm). **h** Dot plots of H2AK119ub average nuclear fluorescence for DMSO− (red) and PTC596-treated (blue) A549 cells. Error bars represent SD of the average values. Data are expressed in fluorescence counts which are proportional to H2AK119ub concentration. *P* value is indicated.

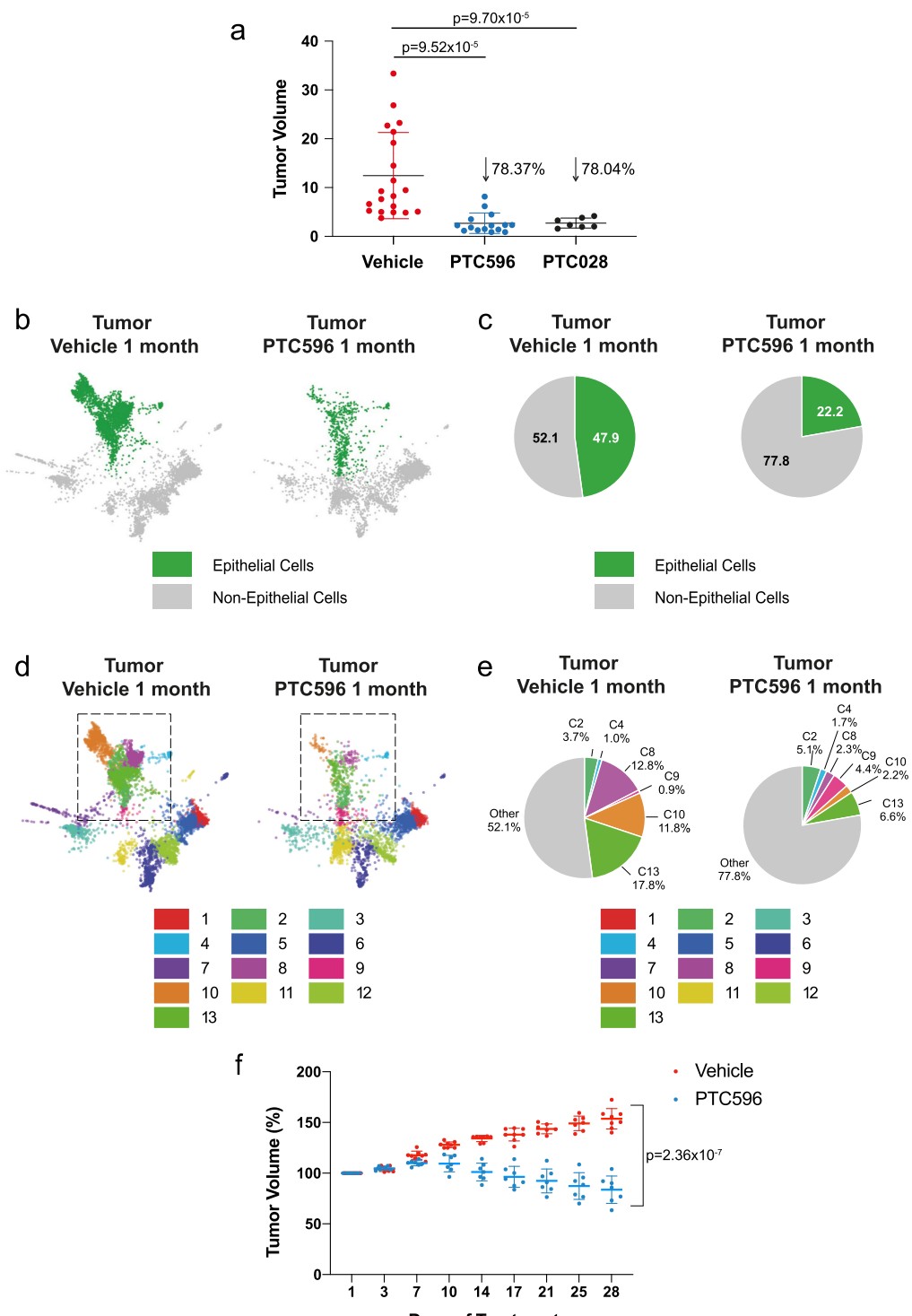

**Fig. 7 PTC596 in vivo treatment affects the growth of _KRAS_-mutant ADCs. a** A549 xenografts tumor volumes at treatment termination (vehicle $n = 20$, red; PTC596 $n = 15$, blue; PTC208 $n = 7$, black) normalized to the tumor volume measured at the beginning of treatment. The difference in tumor size at day 21 was statistically significant ($p$ values are indicated). **b** SPRING plots showing epithelial compartments (green) in vehicle- ($n = 3$, left panels) versus PTC596-treated tumors ($n = 2$, right panels). **c** Pie charts summarizing the epithelial cell proportions in vehicle- (left panels) and PTC596-treated tumors (right panels). **d** SPRING plots of the 13 clusters in vehicle- (left panel) and PTC596- (right panel) treated tumor samples. **e** Pie charts summarizing epithelial sub-cluster compositions in vehicle- (left panels) and PTC596- (right panels) treated tumors. **f** The graph shows the percentage of change in tumor volume measured by MRI at the indicated time points, between PTC596 ($n = 7$, blue) and vehicle-treated ($n = 8$, red) groups. Error bars represent SD. $P$ value is indicated.

decrease in average tumor size by 78.37% ($p = 9.52 \times 10^{-5}$) and 78.04% ($p = 9.70 \times 10^{-5}$) respectively, as compared to vehicle-treated mice.

Having proved these pharmacological treatments significantly affect tumor growth of human mutant-*KRAS* cells in vivo, we tested the clinical-grade PTC596 efficacy in the KP model of lung cancer. Here, we adopted a combined strategy by both profiling lung tumor response to PTC596 through sc-RNAseq and, concomitantly, monitoring in vivo tumor growth by bi-weekly MRI. When tumors reached a comparable size (~1–2 mm diameter), we enrolled the mice for PTC596 or vehicle treatment and followed their response for up to one month. At treatment completion, tumors were isolated and profiled by sc-RNAseq. SPRING plots displayed major differences in the epithelial subpopulations of PTC596- versus vehicle-treated mice (Fig. 7b).

Specifically, epithelial cells represent 47.9% of the vehicle-treated tumor cells, whereas they display an approximately twofold decrease (FDR = $5.75 \times 10^{-57}$), reaching a total of 22.2% overall cells, upon PTC596 treatment (Fig. 7c). SPRING plots in Fig. 7d depict cluster distribution in the vehicle- and PTC596-treated tumors. In particular, ATII-like clusters TEC-C8 and TEC-C13 showed a 5.6- and 2.7-fold reduction, respectively, decreasing their total cell number from 12.8% and 17.8% in vehicle-treated tumors to 2.3% and 6.6% in PTC596-treated tumors, respectively (TEC-C8 FDR = $3.28 \times 10^{-74}$ and TEC-C13 FDR = $2.76 \times 10^{-54}$) (Fig. 7e). TEC-C10, the epithelial tumor-specific cluster, showed a 5.4-fold reduction (from 11.8% to 2.2%, FDR = $3.96 \times 10^{-67}$) (Fig. 7e).

Consistently, MRI showed a concomitant significant decrease in tumor volume (Fig. 7f). Changes in lung tumor burden over the course of treatment were calculated as a percentage change in volume over tumor volume at day 1 (set at 100%). In PTC596-treated mice, tumor size started being significantly impaired by day 7, with notable tumor size regression after day 10. On day 28 of PTC596 treatment, tumors shrunk by 16.3%, as compared to day 1. In addition, although tumor volume increased up to 156.1% by day 28 in the vehicle-treated group, drug-treated tumors shrunk to 83.7%, resulting in a significant 46.3% reduction in tumor size (FDR = $2.36 \times 10^{-7}$), as also shown in the representative MRI images of tumors at day 1 and 28 of both vehicle and PTC596 treatment (Supplementary Figure 5a). Overall, we show that PTC596 significantly decreases tumor size by MRI, in accordance with the major decrease detected in epithelial clusters, as assessed by sc-RNAseq. We also asked whether adopting selumetinib (an allosteric MEK1/2 inhibitor) in combination with PTC596 may show improved antitumor response. Remarkably, MRI data show that PTC596 alone achieves better results than selumetinib alone, and that their combined inhibition is not synergistic, in that adding selumetinib does not significantly improve the effects achieved by PTC596 alone (Supplementary Figure 5b).

Taken together, our data show that BMI-1 inhibition alone is better than MEK inhibition alone and that PTC596 is able to significantly affect murine mutant *KRAS* tumorigenesis in vivo, and almost completely abrogates (5.4-fold reduction, decreasing the subpopulation from 11.8% to 2.2%, FDR = $3.96 \times 10^{-67}$), the murine C10 subpopulation, which shares a common signature with the human mutant-*KRAS*-specific C10 subpopulation, as demonstrated by high-resolution transcriptomics.

## Discussion

Considerable heterogeneity among ADCs of the lung exists, which hampers efficient therapeutic targeting. There remains much to be explored with respect to the physiological significance of molecular and cellular heterogeneity, in an attempt to design improved therapeutic treatments. Therefore, to study tumor milieus at the single-cell transcriptional level, we adopted the inDrop single-cell capture[50], coupled with our newly designed annotation approach, which combines different annotation data sets, therefore allowing for more accurate and robust cell identification. By deconvoluting tumor milieus in both human NSCLCs and a murine model of *Kras*-driven lung cancer, we classified, in both species, clusters/subpopulations comprising the tumor microenvironment. Specifically, we identified endothelial cells, immune infiltrating cells, fibroblasts, and the epithelial component, which is the primary focus of our investigation, as it is a frequent target of transformation[51]. We herein report exceptional parallelism, detected at single-cell level, between human and murine mutant-*KRAS*-driven pulmonary ADCs, which are associated with poor prognosis and aggressiveness. Despite the increased intra-tumor heterogeneity of human ADC samples over clonally bred mice, we discovered a novel tumor epithelial subpopulation (referred to as C10 in the text), which was specifically found in human mutant-*KRAS* ADCs and KP tumors and virtually undetectable in human WT-*KRAS* ADCs and in murine healthy lungs. This subpopulation expresses a set of genes that we identified as a mutant-KRAS-associated signature. Of note, we observed that the mutant-KRAS-associated signature identified in pulmonary ADCs, is also predictive of *KRAS* mutation status in other ADCs (colon, pancreas, and stomach). These data suggest the signature contains oncogenic KRAS-signaling components that act independently of the specific cancer contexts, and that aggressive *KRAS*-mutant-driven malignancies may be similarly targeted.

Despite the current lack of unique markers to identify aggressive pulmonary subpopulations, we discovered a unique molecular signature associated with malignant C10 mutant-*KRAS* cells, which show downregulation of *SCGB3A1* (marker of airway secretory cells), *SFTPA1*, *SFTPC*, *NAPSA*, *SLC34A2*, *LPCAT1*, and *LAMP3* (markers of ATII cells), as well as *HOPX* (the main marker of ATI cells)[42,43], therefore confirming that such population does not match any of the other common epithelial cell type signatures. Furthermore, genes upregulated in the mutant-*KRAS*-specific signature contained oncogenes involved in many human cancers, whose expression is often associated with poor prognosis (Human Protein Atlas, www.proteinatlas.org[37] and[38–41]) and/or resistance to therapy, such as the membrane protein *Plac8*. Noteworthy, its upregulation causes NSCLC resistance to the tyrosine-kinase inhibitor Osimertinib[52] and its silencing in renal cell carcinomas significantly increases their sensitivity to Cisplatin[53].

We also observed that human C10, which is specific to KRAS mutant patients, is present in both TP53 mutant and TP53 wt patients, suggesting the mutant-KRAS-associated signature is independent of TP53-status. Besides having identified a mutant *KRAS*-specific signature, our results show at the single-cell level a model in which KP ADCs are enriched in ATII-like cells, supporting a model in which ATII cells are the cells-of-origin of pulmonary ADCs[54,55]. Tumor-associated epithelial ATII-like clusters and the epithelial cluster specific to mutant-*KRAS* ADCs, are positive for the oncogene *BMI-1*, and, according to our hypothesis, responsive to PTC596 in vivo treatment. Our data represent the first high-resolution analysis of tumor epithelial cells after a targeted pharmacological treatment, proving that sc-RNAseq is a valuable tool to evaluate drug response and assessment of subpopulation dynamics consequent to drug treatment. Indeed, our sc-RNAseq analysis has highlighted an overall approximately twofold decrease in the tumor epithelial compartment size, consistent with the ~50% decrease in tumor volume, as compared to the vehicle, assessed by concurrently monitoring tumor response by MRI.

GSEA of murine C10 cells versus all other tumor epithelial cells demonstrated enrichment of stem cell signatures and target genes of the cancer stem cell gene BMI-1. Consistently, RNA velocity analysis, a bioinformatic tool adopted to elucidate the transcriptional dynamics of cells' evolution, showed that C10 contains a point of origin able to give rise to the other tumor epithelial cells contained within C10, thus corroborating the hypothesis that C10 might contain tumor-initiating cells.

These observations make a significant contribution to the already existing wealth of data proving the relevance of murine models as valid preclinical tools. In this study, we demonstrated their applicability in testing actionable therapies. Overall, our data suggest that large-scale single-cell transcriptomics will eventually impact the development and implementation of enrollment criteria for clinical trials and the evaluation of therapeutic response at the molecular level.

Taken together, our data emerge as a prototype cancer-focused study in which we first identified tumor-specific subpopulations and concomitantly assessed their transcriptional rewiring during target therapy while also following-up tumor growth by MRI. To date, besides the new KRAS-G12C inhibitors[5], trials addressing NSCLC patients harboring *KRAS* mutations have unfortunately not led to major positive therapies[56]. It is noteworthy that PTC596, which is undergoing clinical trials for solid tumors, achieved the therapeutic outcome, we reported with undetectable/unnoticeable side effects. BMI-1 inhibition alone was able to achieve better results than MEK inhibition, and additional pathways will have to be further tested to identify more efficient combinations. Overall, our data strongly suggest that PTC596 might be a promising drug to be tested in the ~30% NSCLC patients carrying highly aggressive *KRAS* mutations.

## Methods

**Cell culture**. The human lung ADC cell lines A549 and SKLU1 were purchased from ATCC. Cells were cultured in Roswell Park Memorial Institute Medium (RPMI) 1640 medium containing 10% fetal bovine serum (FBS, Sigma) and grown at 37 °C in a humidified incubator with 5% $CO_2$. These cells were authenticated via DNA fingerprinting[57] and tested negative for mycoplasma.

Cells line were treated with PTC596 (1 μM), PTC028 (1 μM), or vehicle (0.5% DMSO) for different time points (24, 48, and 72 hours) and the corresponding lysates subjected to Western Blotting to assess BMI-1 protein levels.

**Western blot**. A549 and SKLU1 cells were collected and then lysed with TritonX 1× and cOmplete ethylenediaminetetraacetic acid-free Protein Cocktail (Roche #11873580001) for protease inhibitors and PhosSTOP (Roche #4906845001) as phosphatase inhibitor cocktail. Lysates were centrifuged at 12,000 × g for 15 min at 4 °C, the supernatant was snap frozen in Liquid $N_2$ and stored at −80 °C. In all, 15 μg of total protein were separated on 10% sodium dodecyl sulfate-polyacrylamide gel electrophoresis gels and transferred to nitrocellulose membrane using the TransBlot Transfer System (BioRad). Membranes were blocked in tris-buffered saline and tween 20 containing 5% non-fat dry milk and incubated with primary antibodies anti-BMI-1 (1:1000 Cell Signaling Technologies #6964 S). Membranes were then stripped with Restore Western Blotting Buffer Solution (Thermo Fisher #21059) for 15 minutes at room temperature (RT) and incubated overnight (O/N) with an anti-β-actin mouse antibody (Santa Cruz #81178) at a 1:1000 dilution to assess equal loading. Blots were incubated with specific HRP-conjugated secondary antibodies, anti-rabbit IgG-HRP (Santa Cruz #SC2054) or anti-mouse-IgG-HRP (Santa Cruz #SC2031). An enhanced chemiluminescence blotting analysis system (Pierce, Thermo Scientific #32106) was used for antigen-antibody detection. The density of western blot bands was quantified by ImageJ software (Version 1.51m9, National Institutes of Health, Bethesda, MD, USA).

**Cell cycle analysis**. A combination of Vybrant DyeCycle Violet and Pyronin Y was used for the differential staining of cellular DNA and RNA. A549 and SKLU1 cells treated for 24 hours with PTC596, PTC028, and DMSO, were permeabilized in phosphate-citrate buffer solution (pH 4.8), washed in phosphate-buffered saline (PBS) 1×, and then resuspended in a solution of 5 μM Vybrant DyeCycle Violet (Thermo fisher Scientific) and 4 μg/ml pyronin Y (Polysciences). Cycle status was then evaluated by flow cytometry on Cytoflex Flow Cytometer (Beckman Coulter, Inc.).

**Immunofluorescence analyses on A549 cells**. A549 cells were washed with phosphate buffer saline 1× (PBS, three times) and then fixed with paraformaldehyde (2% in PBS) for 15 min. After washing with PBS (three times), cells were permeabilized with 0.1% TritonX-100 made in PBS, for 15 min. Cells were then washed with PBS (three times), 0.5% bovine serum albumin in PBS (PBB) four times), and exposed for 40 min to 2% BSA in PBS (BSA 2%). After washing with PBB (four times), cells were incubated with rabbit anti-human ubiquityl-histone H2A monoclonal antibody (Cell Signaling Technologies #8240 S; 1:1600 dilution in PBB) for 1 h at RT and additional 1.5 hours at 4 °C. Cells were washed with PBB (four times), and incubated with the secondary antibody (donkey anti-Rabbit IgG AlexaFluor 647 from Jackson ImmunoResearch #711-605-152 at 1:250 dilution in PBB) for 1 h at RT in dark. Next, cells were washed with PBB (four times), stained with Hoechst 33342 (1 mg/100 ml in water) for 30 s, and washed with PBS (three times). Cells were then maintained in PBS at 4 °C before imaging, no longer than 7 days.

The negative control was obtained by means of the same procedure but incubating the cells with PBB only instead of a primary antibody solution in PBB.

Imaging was carried out on a Zeiss 880 LSM confocal microscope according to the protocol reported in Storti et al.[49].

**Murine models**. Mice were housed in a sterile-barrier facility, and all experiments were approved by the Institutional Animal Care and Use Committee at the Beth Israel Deaconess Medical Center.

**Xenografts and drug treatment**. To study the in vivo effect of PTC596 or its analog PTC028 on the ADC cell line A549, *NOD-SCID IL2Rγ(null)* mice (non-obese diabetic/severe combined immunodeficient/interleukin-2 receptor γ null, NSG mice, Jackson Laboratories) were injected subcutaneously in flanks on both sides with $2 \times 10^6$ cells with 50 μl Matrigel (BD Basement Membrane Matrix Phenol-red free #356237). Once tumors became measurable (~80–90 mm³), mice were randomized to receive PTC596 (n = 15; 12 mg/kg), PTC028 (n = 7; 15 mg/kg), or vehicle (n = 20; 0.5% hydroxypropyl methylcellulose—0.2% Tween 80 in distilled water) by oral gavage twice a week. In order to determine tumor volume by caliper measurement, the greatest longitudinal diameter (length) and the greatest transverse diameter (width) were determined. Tumor volume was calculated by the modified ellipsoidal formula (tumor volume = ½ (Length×width²), as previously reported[31]. Treatment was started when tumor volume was measured with a caliper as being at least 0.06 cm³, and tumor growth was followed up to 21 days.

**Transgenic mice, drug treatments, and MRI**. To generate *K-RasG12D/p53* null mice (Lox-stop-lox/LSL x K-RasG12D, p53 flox[18,19,58] they received intranasal administration of Cre-expressing adenovirus ($2.5 \times 10^7$ viral particles per mouse)[58] at 8 weeks of age, to achieve recombination in airway cells[19]. Sibling mice, receiving the same amount of Adeno-empty virus, have been used as negative controls to study healthy tissues. Tumor growth was assessed by MRI at the BIDMC Imaging Facility after 5–6 weeks of induction, and were then monitored every 1–2 weeks to detect baseline tumor volume and recruited into treatment groups (PTC596, or vehicle) when tumor size reached 1–2.0 mm diameter.

Mice were treated with PTC596 (n = 7; 12 mg/kg in 0.5% HPMC and 0.2% Tween, oral gavage twice per week), selumetinib (n = 12; daily oral gavage, 8 mg/kg in 0.5% HPMC and 0.2% Tween, oral gavage daily), combination of PTC596-selumetinib (n = 6) or vehicle control (n = 8; 0.5% HPMC and 0.2% Tween, same regimen as PTC596). Negative control mice received vehicle treatment. All mice were killed 1 month after treatment when the tumor burden of vehicle-treated mice became too large.

Mice were then scanned by MRI twice/week to capture the effects of drug treatment on tumor size over a month period. Processing and quantification techniques of tumor burden were based on manual segmentation/volume calculation of diffuse lung tumors[59,60]. Changes in lung tumor volumes over the course of treatment were calculated as a percentage change in volume over tumor volume at day 1 of treatment, which was set at 100%. MRI images of mouse lungs were captured with a Bruker Biospec 94/20 9.4 Tesla scanner and the primary imaging sequence used was RARE (rapid acquisition with refocused echoes), with TR/TE = 1200 ms/17.5 ms.

**Histopathological analyses**. Mice were killed by $CO_2$ euthanasia. Lungs and xenograft subcutaneous tumors were fixed in 10% formalin (formalin solution neutral buffered 10%, Sigma-Aldrich) O/N. Fixed specimens were embedded in paraffin and sectioned at 5-μm thickness. IHC were performed on paraffin sections with an anti-BMI-1 mouse monoclonal antibody (Millipore, #05637; 1/100 dilution) on mouse tissues, and a rabbit anti-human BMI-1 (Cell Signaling #6964; 1/200 dilution) for the xenografts. Fresh tumor tissue was collected from patients undergoing surgical resection of NSCLC and placed in RPMI prior to being fixed in 10% formalin. Subsequently, they were stained as mentioned above. In brief, tissue sections were deparaffinized with xylene and hydrated in graded ethanols. Antigen retrieval was performed in 10 mM citrate buffer (pH 6.0) on a 2100 Retriever for 40 min. To prevent non-specific binding we applied as protein blocking solution 7% horse serum in PBS for 30 minutes at RT. Primary antibody were incubated at

4 °C O/N. Next, we applied peroxidase blocking solution for 10 minutes at RT and subsequently we performed secondary antibody incubation for one hour at RT. Secondary antibodies were horse anti-mouse BA2001 at 1/1200 dilution, and goat anti-rabbit BA1000 at 1/1000 from Vector Laboratories, Inc. CA. ABC-HRP standard kit (Vector Labs, CA PK-6100) was adopted and incubated for 30 minutes and the signal was revealed with DAB (Vector Labs, CA SK-4100). Tissue sections were counterstained with hematoxylin–eosin and mounted with Cytoseal 60 (Electron Microscopy Science), for pathology analysis.

**Mouse lung and tumor dissociation into single cells.** Murine pulmonary tissue (healthy and tumor lung) were dissociated into single-cell for further RNA sequencing downstream applications using the Tumor Dissociation kit by Miltenyi Biotec (# 130-096-730). In brief, the tissue was placed in a petri dish on ice and cut into small pieces of 2–4 mm. The pieces were infused with RPMI/enzyme mix (Miltenyi Biotec), transferred to a gentleMACS C tube containing RPMI/enzyme mix, attached to the sleeve of the gentleMACS Octo Dissociator and run using a "37C_m_TDK_1" program. After termination of the program, the cells were spun down at 300 × g for 10 min at 4 °C, resuspended in RPMI-2% FBS, passed through a 70 μm strainer and centrifuge was repeated. The cell pellet was treated with 1 ml of ACK solution for 7 min at RT, and the lysis stopped with 4 ml of RPMI-2% FBS. After centrifugation, the cells were suspended in 1 ml RPMI-2% FBS and passed through a cell strainer (70 μm) to obtain a single-cell suspension. Immediately before transcriptome barcoding using the inDrop platform, cells were manually counted on a hemocytometer and diluted to 60,000 cells/ml. The final cell suspension included 15% v/v Optiprep (Sigma-Aldrich, Cat. No. D1556).

**Patient description and preparation.** Twelve NSCLC (twn ADCs and two squamous carcinomas) patient samples were analyzed in this study. Fresh samples were obtained from patients undergoing surgical resection. Patients were selected when having tumors ≥1.6 cm that were treatment-naive. Analysis related to mutant-KRAS versus wt-KRAS mutation status was only performed on ADC samples (n = 10).

This study was conducted with approval of the Dana-Farber Brigham and Women's Cancer Center IRB and written informed consent from subjects. The protocol allows the collection of discarded tissue samples. De-identified genomic information is deposited in protected public repositories for subjects explicitly allowing it on the consent form. Human tissue samples were de-identified and analysis is not considered human subject research under the US Department of Human and Health Services regulations and related guidance (45 CFR part 46). Perpendicular sections immediately flanking 1–3-mm thick fragments of all tumor tissues were reviewed by a pulmonary pathologist to confirm the diagnosis and tumor content. Patients' information is shown in Supplementary Table 1.

Tumor lung samples were dissociated for sc-RNAseq using a Tumor Dissociation kit Human from Miltenyi (#130-095-929), similar to the protocol described for murine tumors, except the "37C_h_TDK2" program was used.

Genomic DNA was extracted from four 10 μm scrolls of paraffin-embedded tissues per sample utilizing the QIAamp DNA FFPE Tissue Kit (#56404), according to the manufacturer's instructions.

All clinical samples were sequenced using the Oncopanel platform[61]. Samples 18, 21, 36, and 37 were sequenced using Ion S5-targeted sequencing (Ion Torrent; Applied Biosystem, Calsbad, CA, USA). KRAS mutation status was confirmed by digital droplet PCR (ddPCR) using the ddPCR Mutation Assay (BioRad®, Hercules, CA) as per manufacturer instructions. A fluorescence intensity threshold of 3000 was set as a cutoff point. A mutation was called when at least one droplet was above the threshold level. Mutations values were reported as mutant allele frequency (MAF), defined as the proportion of mutant to wild-type PCR products in the ddPCR readout. The analyzed mutations were KRAS codons 12 and 13. Patients have been divided into KRAS mutated and KRAS wt (Supplementary Table 1) based on the presence of activating mutations (see Supplementary Data). Some tumors displayed KRAS amplification or loss. However, no functional analyses were conducted to determine the effects of these additional mutations.

**InDrop.** For inDrops-seq, the cells were encapsulated in droplets and the libraries were made at the Harvard Single-Cell Core[50,62] with the following modifications in the primer sequences.

RT primers on hydrogel beads-
5′-CGATTGATCAACGTAATACGACTCACTATAGGGTGTCGGGTGCAG
[bc1,8nt]GTCTCGTGGGCTCGGAGATGTGTATAAGAGACAG[bc2,8nt]
NNNNNNTTTTTTTTTTTTTTTTTTTTTV-3′

R1-N6 primer sequence (step 151 in the library prep protocol in [2])- 5′-TCGT
CGGCAGCGTCAGATGTGTATAAGAGACAGNNNNNNN-3′

PCR primer sequences (steps 157 and 160 in the library prep protocol in [2])-
5′-AATGATACGGCGACCACCGAGATCTACACXXXXXXXXXTCGTCGG
CAGCGTC-3′, where XXXXXX is an index sequence for multiplexing libraries.
5′-CAAGCAGAAGACGGCATACGAGATGGGTGTCGGGTGCAG-3′-

**sc-RNAseq data processing, quality control, filtering, and cell type identification.** Raw fastq files were obtained from a NextSeq 500. Next, transcripts were mapped to the mouse transcriptome (GRCm38.81) following an established bioinformatics pipeline for inDrop experiments to create raw gene-cell counts

matrices (available on GEO: GSE136246)[50], with the slight modification of adding an additional filtering step to remove lower quality base calls using Trimmomatic (v0.36)[63] with the parameters "LEADING:28 SLIDINGWINDOW:4:26 MIN-LEN:15" after the "identifying abundant barcodes" step. Human samples were mapped to GRCh38.91 using a similar approach. Subsequent analysis was performed in R (v3.6.2) and was based on a previously published Bioconductor workflow with minor modifications[64], and published guidelines for sc-RNAseq analysis[65]. In brief, cells with low total raw counts were removed if they had fewer than a calculated number of counts. This trim threshold was based on the mode of the total counts for a given sample and was calculated as follows (eq. 1).

$$
\begin{cases}
\text{If mode\_estimate} < 100, \text{trim\_threshold} = \text{mode\_estimate} * 5.5 \\
\text{If } 100 \leq \text{mode estimate} \leq 450, \text{trim\_thresold} = (-0.01 * \text{mode\_estimate}) + 6.5 \\
\text{If mode\_estimate} > 450, \text{trim\_threshold} = \text{mode\_estimate} * 2
\end{cases}
$$

(1)

Next, cells with library sizes more than three median absolute deviations (MADs) below the median library or six MAD's above the median library size were filtered out. Cells with a total number of expressed genes (≥1 read) more than two MADs below the median total number of expressed genes or five MAD's above the median total number of expressed genes were filtered out. Cells with a total percent of expressed genes originating from mitochondrial DNA more than six MADs above the median were filtered out. A doublet score was computed to estimate the percentage of barcodes for two or more cells[66], and cells with a doublet score of 0.99 and greater were excluded. The expression of each cell was normalized by a size factor approach[67]. Principle component analysis, Uniform Manifold Approximation and Projection for Dimension Reduction and t-distributed stochastic neighbor embedding visualizations revealed no significant batch effects to be regressed out for the mouse samples. For the human samples, mnnCorrect() from the batchelor package[68] was utilized for batch correcting samples.

For expression data visualization, SPRING was used[69]. In brief, a graph of cells connected to their nearest neighbors in gene expression space was determined and this was then projected into two dimensions using a force-directed graph layout. For cell annotations for the murine samples, a custom hybrid annotation was used wherein immune cells were labelled using SingleR[33,35,70] and its Immgen reference set, whereas non-immune cells were labeled using the Mouse Cell Atlas as a reference[71]. For the human samples the SingleR Human Cell Atlas annotation was used[11].

Supplementary Table 9 shows a cross-table of (a) predominant cell populations present in each murine and/or human cluster and (b) cell types across both species.

For the patient samples, a total of 23 tumor nodules were sequenced with the number of nodules from each patient as shown in Table 1:

Throughout the paper, we refer to deconvolution that was performed considering solely the ADC samples contained within the NSCLC set and the mutant-KRAS-associated cluster was annotated as Cluster 10 (C10). For validation purposes, we also deconvoluted only the ADC samples, and in such case, the mutant-KRAS-associated cluster was annotated as Cluster 3 (C3), as shown in Supplementary Figures 1b, c and Supplementary Figure 3a.

**Differential gene expression analysis, marker gene identification, and figure generation.** Differential gene expression was calculated using genewise negative binomial generalized linear models (glmFit()) from the edgeR package[72,73]. In brief, binomial dispersion was estimated using the estimateDisp(), followed by model fitting using glmFit() and finally, likelihood ratios for differential expression were calculated using glmLRT(). Differential gene expression results are provided in Supplementary Tables including Log2 fold change (Log2FC), p value and Benjamini–Hochberg adjusted p values (FDR)[74].

For the heatmaps in Supplementary Figure 1, heatmaps were generated as follows. To identify marker genes per expression cluster, findMarkers() from the scran package[64] was utilized with the parameter (direction = "up" or "any")

**Table 1 Patients' sample replicates.**

| Patient sample | Number of replicates |
|---|---|
| NSC004 | 4 |
| NSC009 | 2 |
| NSC010 | 2 |
| NSC016 | 3 |
| NSC018 | 2 |
| NSC019 | 2 |
| NSC020 | 2 |
| NSC021 | 2 |
| NSC035 | 1 |
| NSC036 | 1 |
| NSC037 | 1 |
| NSC040 | 1 |

depending on the heatmap created. From the 25 highest expressed genes per cluster in tissue samples (25 genes, 13 cluster, totaling 325 genes), only unique genes were selected (i.e., genes identified as a marker in only 1 cluster) and shown. This resulted in a final count of 235 genes for murine healthy samples and 222 for murine tumor tissue samples. Heatmaps were generated using the pheatmap R package (Kolde R. pheatmap: Pretty Heatmaps. R package version 1012 2019) with normalized expression matrices being row (gene) scaled prior to plotting. Genes (rows) were hierarchically clustered in an unsupervised manner based on Euclidean distances between genes by toggling the flag "cluster_rows=TRUE". The genes corresponding to the rows of the heatmaps are listed in Tables S3 and S4, with genes in the same order as the heatmap.

**Ingenuity pathway analysis**. Analysis was conducted using IPA (Ingenuity® Systems, www.ingenuity.com). Gene lists containing mean expression values of sc-RNAseq data of murine and human C10, filtered for $p$ value $\leq 10^{-10}$, were submitted to IPA. Core analysis of the uploaded data identified canonical pathways, diseases and functions, and gene networks that are affected. Heatmaps were used to show comparison analysis between murine and human data.

**Gene set enrichment analysis**. Differential expression gene sets were filtered with an FDR < 0.01 and then ranked based on logFC. GSEA was then performed by comparisons with curated expression grp lists downloaded from the Broad Institute using the GSEA tool also from the Broad. Specifically, we used murine data sets M19097[75], M9118[76], M8795[77], M1999[78], and M9473[79] and human data sets M5953, M2897, M27039 M4572[80], M2534[81], M7079[78], M1260[82], M16956[83], and M2316[84]. For the Bmi-1 targets, we used the murine data set GSE56935[31]. The GSEA data set database used was the full version of MSigDB v7.2.

**Survival analysis and mutation-level association analysis**. For survival analysis of C10 signature, lung adenocarcinoma samples were taken from TCGA (https://tcga-data.nci.nih.gov/tcga/), and the RNAseq raw counts data and clinical data from the data sets were utilized. Prior to survival analysis, TCGA RNAseq raw counts were normalized using the total mappable reads across all samples. Then, $z$ score transformation was performed for each gene in the C10 signature cross all samples. For each of downregulated genes in C10, the $z$ score was further transferred by multiplying with $-1$. The overall C10 score for a sample was determined by averaging the $z$ scores of all upregulated genes and the transferred $z$ scores of all downregulated genes. The median of all overall C10 scores across all samples was used to segregate the samples into C10 signature high and low groups. Survival analysis was performed between these two groups (C10 signature high— higher than the overall score median and C10 signature low—lower than the overall score median) based on the Kaplan–Meier method using overall survival data.

TCGA ADC cohorts (colon, COAD; pancreas, PAAD; stomach, STAD) were scanned with the C10 to score its enrichment in KRAS-mutant versus WT cancers.

**EnrichR analysis**. Following differential gene expression of C10 versus other epithelial clusters in both murine and human data sets, gene lists, filtered LogFC $\geq 1.5$, FDR $\leq 0.05$, were submitted to the EnrichR enrichment suite[85,86].

**Method for scoring the signature genes**. TCGA RNAseq samples from several ADCs (colon, rectum, pancreatic, stomach) were downloaded for gene expression analysis. After mapping with STAR software, mapped counts were employed to generate the raw expression counts using the FeatureCounts. The raw expression counts were then further normalized using the Cross-Correlation method[87]. *Kras* mutation status (mutant or wild type) across all samples was evaluated based on the called maf files. For each signature gene in a sample, $z$ score transform of normalized expression was performed over all samples in each ADC type. The $z$ score transferred values of all signature genes in a sample were averaged to obtain the signature score of the sample. For a combination of the positive and negative *Kras*-mutant signature genes, negative equal weights were given for all negative signature genes and the same positive equal weights for all positive signature genes prior to averaging.

Student $t$ statistics were used to assess the significance between the two means of the signature scores in *Kras*-mutant and WT sample groups.

**Statistics and reproducibility**. For comparison of continuous variables between groups, we used $T$ test (two-tailed; type 3) unless otherwise stated. Differences were considered statistically significant at $P<0.05$. The association between categorical variables was investigated with two-sided Fisher's exact test on cell numbers per cluster. FDR's were calculated by adjusting $p$ values using the Benjamini–Hochberg method. Statistical analyses were performed in PASW Statistics 18 (SPSS Inc.) and R version 3.6.1 (The R Foundation for Statistical Computing) at 5% significance level.

**Reporting summary**. Further information on research design is available in the Nature Research Reporting Summary linked to this article.

## Data availability
The data sets generated during and/or analyzed during the current study are available in the GEO repository: GSE136246. All other data are available from the corresponding author on reasonable request.

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

## Acknowledgements

This work is funded by BIDMC Chief Academic Award (CAO 2015) to E.L., BIDMC Jax Pilot Award (BIDMC/Jax) to E.L., Yong Siew Research Grant from National University of Singapore WBS R713-001-013-271 to E.L., Regione Toscana Bando FAS Salute 2014 (to E.L. and R.B.), MIUR (Ministry of Education, University and Research) Flagship Interomics Project 20/2017 to E.L, sponsored research support from PTC Therapeutics to E.L., the National Research Foundation Singapore under its Singapore Translational Research (STaR) Investigator Award (NMRC/STaR/018/2013) to D.G.T.; the Singapore Ministry of Health's National Medical Research Council (Open Fund Large Collaborative Grant LCG17MAY004) to D.G.T.; the National Research Foundation Singapore and the Singapore Ministry of Education under its Research Centres of Excellence initiative to D.G.T.; the Singapore Ministry of Education Academic Research Fund Tier 3, grant number MOE2014-T3-1-006 to D.G.T.; and NIH grants R35CA197697, 5P01CA66996, and P01HL131477 to D.G.T. and R.S.W., R713-000-216-720 to A.A., NIH grants R33-CA212697 and R01-CA218579 to A.M.K., Grant #2016/19757-2 Fundação de Apoio à Pesquisa do Estado de São Paulo (FAPESP) to D.S.B., MIUR Flagship InterOmics Project RBNEO01R4MJ-002 to M.C.M., startup funds from the Division of Hematology/Oncology at the University of Alabama at Birmingham (UAB) Bridge Grant (2018) and the American Society of Hematology to R.S.W. R.B. (NEST) acknowledges the funding from the Short Term Mobility 2018/2019 of the National Research Council of Italy (CNR). We thank the Harvard Single-Cell Core, the Bauer Core Facility, the BIDMC Preclinical Murine Pharmacogenetics Core, PTC Therapeutics (South Plainfield, NJ) for generously donating the BMI-1 inhibitors PTC596 and PTC028, and the DFCI Oncology Data Retrieval System (OncDRS) for the aggregation, management, and delivery of the clinical and operational research data used in this project. The content is solely the responsibility of the authors. We also acknowledge Pier Paolo Pandolfi De Rinaldis, M.D., PhD (HMS), William G Richards, PhD, Director of the Tissue and Blood Repository core of BWH, and Daniel B. Costa, M.D. (BIDMC) for helpful discussion.

## Author contributions

E.L., D.G.T., A.A., G.M., I.K., and M.A.B. designed the study; E.L., A.A., G.M., I.K., M.A.B., J.C., D.S.B., A.G., M.D.R., RB, R.S.W., and J.G.C. performed and planned research; E.L., D.G.T., A.A., G.M., I.K., M.A.B., V.S., R.Z., B.S., J.C., R.P., J.L., D.S.B., M.W., A.B., A.G., R.C., M.D.R., R.D., R.B., H.Y., O.K., M.C.M., R.S.W., A.M.K., R.B., and J.G.C. analyzed data; F.C.W., V.S., R.Z., V.A.M., N.P., E.C., J.Z., C.E.G, S.F., R.D.L., C.V.M., P.J.T., K.V., A.D.R., S.C., and R.C. performed research; and E.L., D.G.T., A.A., G.M., I.K., and M.A.B. wrote the paper.

## Competing interests

The authors declare no competing interests.
