## [Peer Review File · Communications Biology]

Reviewers' comments:

NOTE: Reviewer #1 and #2 reviewed this manuscript together. Their shared comments appear below.

Reviewer #1 and #2 (Remarks to the Author):

COMMSBIO-20-0667-T

Maroni et al. "Identification of a targetable KRAS-mutant epithelial population in Non-Small Cell Lung Cancer "

The authors perform scRNAseq analyses on 12 non-small cell lung cancer patients (2 squamous and 10 adenocarcinomas) and identify 15 different cell clusters. Of these cluster 10 were thought to be tumor cells in the KRAS mutant tumors. They also performed scRNAseq on tumors from genetically engineered mice with KRAS and TP53 mutations ("KP" mice) and found 13 clusters. They performed computational tests to relate the mouse and the human cluster. They go onto show that BMI-1 is upregulated in the KRAS tumor cell cluster with an associated mRNA expression signature. They perform in vitro and in vivo studies of an oral BMI-1 inhibitor (PTC596) that is in clinical trials and find loss of BMI-1 expression and inhibition of tumor growth in human A549 xenografts and mouse syngeneic tumors. As part of this they also performed scRNA seq on tumor material after PTC596 treatment. A variety of computational analyses of deposited databases are presented combining their signature using mRNA expression across KRAS mutant solid tumors. No information on the general mutation status of the human tumors they studied, including tumor mutation burden, or immunohistochemical validation of BMI-1 or any of the marker gene signatures on human tumor samples were presented.

Comments to the authors:

This paper is reviewed in the context of the urgent need to have targeted therapy for KRAS mutant lung tumors with associated biomarkers to allow precision medicine. It is also reviewed in the context of many papers reporting scRNAseq data on relatively small numbers of patients (like this report) with non-small cell lung cancers and trying, like the current study, to use this information to both understand the pathogenesis of non small cell lung cancer, and to identify new targets for therapy particularly in KRAS mutant NSCLCs. While the study was technically well done, it was presented in a complex and unwieldy manner that made it hard to follow. The authors need to address several issues:

1. They need to focus on the 10 lung adenocarcinomas since squamous cancers almost never have KRAS mutations. The two squamous patients do not add any thing to the analyses. In addition, they need to have their text and patient table correlate – they have 7 KRAS mutant and 3 wild type (not 2 as stated) LUADs. There were several other examples, where their complex writing style led to having to "track things down" which was not desirable to the reader.
2. They need to provide simple demographics about smoking status of the 10 LUADs they studied.
3. They found 15 different clusters in humans and 13 clusters in mouse materials. We need to know in a simple manner how did each of the human clusters map to (or not map to) a mouse cluster. We are left thoroughly confused about this simple fact.
4. It appears that human cluster 10 was the key tumor cluster and there was a mouse cluster that mapped to this. If there are other human clusters that contain tumor cells we need to know and did they map to any mouse clusters? Also we know there can be intra-tumor heterogeneity (ITH) in LUAD – was there such ITH found here, and was this represented by Cluster 10 and some other tumor clusters? Again, this simple fact is not easily discernible.
5. There is discussion of 21 up and 14 down regulated genes presumably in Cluster 10. But then we hear about 50 down regulated genes. Which is correct?
6. All of their preclinical studies were done with the old "war horse" for NSCLC studies A549 cells. Of course A549 is wild type for TP53 (while the GEMM models are TP53 and KRAS mutant) making them not comparable. In addition, A549 cells are STK11 mutant, which has major implications for tumor behavior. None of these aspects were controlled. Of course, the usual things would be to have several xenograft models, and potentially patient derived xenograft (PDX) models that were tested.

7. The xenograft studies were done on very small tumors (they report starting treatment with sic "2 to 3 mm" diameter tumors). While there is nothing "wrong" with this, typically one would start treatment on larger size tumors.

8. It is very hard to know how much of an anti-tumor effect the PTC596 treatment had. They use words like "virtual complete disappearance" of a tumor cell population by scRNAseq analyses. But looking at Figure 5 f. the % tumor volume goes from 100 to 80%. Also what does "tumor volume" on the y-axis of figure 5 a mean?

9. It would have been important to know the mutation status of many genes they would have from their RNAseq data to find out about other key mutations (e.g. TP53, STK11 to name a few) and also the tumor mutation burden, and expression of PD1, PDL-1 on tumor cells and in the tumor microenvironment. The reasons for these studies are obvious given both the widespread use of CLIA certified mutation testing on NSCLC, and the movement of checkpoint inhibitor therapies into first line treatment of NSCLCs.

10. Since BMI-1 is a key cancer stem cell expressed gene, it would have been interesting to know if PTC596 treatment actually removed the cancer initiating cells in either clonogenic assays, monitoring other cancer stem cell markers (such as ALDH related genes), or even better in serial transplantation studies. Also whether they did this equally well in KRAS mutant vs. KRAS wild type tumors. This type of information would be important for precision medicine clinical translation.

Reviewer #3 (Remarks to the Author):

This manuscript describes a subset of lung adenocarcinomas with a specific molecular signature. RNA sequencing was performed, and the effects of in vivo PTC596 treatment, which affects BMI-1 activity, was evaluated in the murine model.

Lung cancer is a common disease with a poor prognosis. In adenocarcinomas of the lung, KRAS-mutations are typically found in 1/3 of the tumors. Up to now, there has been efforts in developing treatment specifically targeting KRAS, but no such treatments are implemented in clinical practice yet. However, a very promising drug have shown exquisite results for lung cancer patients with the G12C mutation treated with AMG 510. A mention of this would improve the introduction.

The investigations performed molecular analyses including sequencing, cell lines studies and mouse model work. The analyses give interesting results and seem to be thoroughly done.

Some analyses are performed on tumor tissue from lung cancer patients with KRAS-mutation in their tumor. It would be interesting to know which KRAS-mutation where found, and several papers have reported different biology depending on the different KRAS-mutations found. Were the samples also analyzed for other mutations? How was the tumors selected?

The authors call it a subpopulation, but the signature seems to be present is all the KRAS-tumors, in contrast to normal cells.

It would be very interesting to see the MRI images of the mice, - showing examples of the shrinking. The authors call it shrinking and at the same time 50% reduced tumor growth. Figure 5f shows a reduction in tumor volume. I would then suggest not to call it tumor growth, as one can get the impression that the growth rate is reduced, but it is still growing. Were any mutational analyses performed on the tumors after treatment?

How was the "take" in the mouse model work?

Reviewer #4 (Remarks to the Author):

This manuscript by Maroni et al reports the identification of a mutant K-RAS associated gene signature, which is conserved in both human and murine lung cancers. The authors show that

PTC596, a BMI inhibitor, inhibits tumor growth in xenograft and mutant K-RAS/p53-null (KP) mouse models with concomitant elimination of C10 subpopulation of epithelial cells. As there is no targeted therapy approved for mutant K-RAS-mediated NSCLC, identification of a tumorigenic and targetable subpopulation may have profound impact in designing improved therapies. This is an interesting work and the conclusions are broadly supported by the experimental data provided. The authors have done a good job in verifying new observations, pursuing orthogonal methods to confirm specific results, and drawing meaningful conclusions. Overall, I do believe that this work is of the caliber to be published in this journal. The data presented in this study are clear, well-presented, statistically significant, and adequate controls are included in the experiments. There are some unanswered questions and this study would be further strengthened by addressing the following:

1) The authors have made a key statement that there is a tumor enriched epithelial cluster, C10, which is specific for mutant-KRAS adenocarcinomas (ADCs) compared to WT K-RAS ADCs. In lung ADCs, mutations in K-RAS and p53 co-occurs in 40-45% cases, and p53 mutations takes place in almost double the number of K-RAS mutant cases. This C10 cluster is also present in tumors from combined mutant KRAS and p53-null mouse (KP). Therefore, is it possible that this cluster of epithelial cells is conserved for KRAS and p53 loss-of-function mutations? Similarly, the associated transcriptomic signature should be attributable to both K-RAS and p53 mutations in lung cancer. Along this line, it is better to mention how many of the seven K-RAS mutant patients have also p53 mutations (Supplementary Table 1).

2) Ingenuity Pathway Analysis (IPA) identified few signaling pathways like EIF2, mTOR and eIF4/p70S6K as the top three enriched pathways in human and mouse C10 populations. In addition, IPA detected activation of other pathways including Integrin, EGF, IGF-1, MEK/ERK, and Insulin receptor in the C10 population from human and mouse K-RAS associated lung tumors. The rationale for choosing the oncogene BMI-1 as a target for treating this C10 population of tumor cells is not clear, although the mutant K-RAS ADC cells are positive for it, as suggested by the authors. BMI-1 is not listed in the top marker genes in tumor tissues (Supplementary Table 4) and it is not among the top 50 highest expressing genes in the transcriptional profile C10 (Supplementary Table 7). Therefore, why BMI-1 was chosen as the target for this K-RAS specific cluster, and not other activated pathways or genes that are on the top of the list.

3) It is important to discuss whether BMI-1 is a downstream target of any of those signalings mentioned above, which are activated in the K-RAS specific C10 epithelial population. It is worth mentioning for the differential expression of all of the BMI-1 upstream regulators (like MYC, SALL4, Nrf2, etc) or downstream targets (like Ink4a, Arf, PTEN, Hox, VEGF, etc) in the scRNA-seq data from C10 population with or without the treatment by PTC596. As BMI-1 is a regulator of cancer cell stem phenotype, what is the status of expression of stemness/differentiation markers in this cell population before and after treatment?

4) BMI-1 activates downstream oncogenic functions through its upregulation in the tumor. But here, the authors did not observe appreciable change in its expression in the C10 population, rather other oncogenic pathways like MEK/ERK, mTOR, NF-kB, and p70S6K are strongly activated. Therefore, the C10 population of tumor cells can be best targeted by MEK/ERK, mTOR, and p70S6K inhibitors that are already approved. To clarify which inhibitor targets these cells more efficiently, it is better to compare the efficacy of PTC596 with that of MEK/ERK and p70S6K inhibitors in xenograft assays with lung tumor cell lines. In addition, tumor shrinkage with PTC596 is only 50% in the KP mouse model. It will be interesting to determine if the combination of PTC596 with MEK/ERK, mTOR, NF-kB or p70S6K inhibitor can reduce the tumor growth completely with concomitant abolishment of murine C10 subpopulation. These experiments can be done using xenograft models

5) Although BMI-1 is not upregulated in the human and mouse tumor enriched cluster (TEC) C10, its inhibitor has significant effect on the tumor regression with corresponding reduction of this population. It raises an important question whether PTC596 inhibits any of the activated pathways like mTOR, p70S6K, MEK/ERK, EGF, NF-kB, etc. in the C10 population. These can be shown in lung cancer cell culture studies.

6) It is better to provide the mutational status of important genes in patient lung tumors in the Supplementary Table 1, if available.

6) A brief description of the BMI-1 protein and its function in the 'Introduction' of the manuscript will be helpful.

7) In some figures labeling is very small and illegible. It should be fixed.

ELENA LEVANTINI, Ph.D.

Center for Life Sciences

3 Blackfan Circle
Boston, MA. 02215

Tel: 617-735-2233 • **Fax:** 617-735-2222

E-mail: elevanti@bidmc.harvard.edu

November 6th, 2020

To Reviewers #1-4:

We appreciate that all reviewers recognized the strengths of our manuscript, indicating that i) our work contains “interesting results”; ii) “the conclusions are broadly supported by the experimental data provided”; iii) “the authors have done a good job in verifying new observations, pursuing orthogonal methods to confirm specific results, and drawing meaningful conclusions”; iv) “it is technically well done”, and v) “this work is of the caliber to be published in this journal, as the data presented in this study are clear, well-presented, statistically significant, and adequate controls are included in the experiments”.

There were a number of excellent suggestions for improvement made by the reviewers’ panel that we addressed in every instance, therefore substantially strengthening our manuscript, as indicated below:

Reviewers 1 and 2

Remarks to the Authors:

No information on the general mutation status of the human tumors they studied, including tumor mutation burden, or immunohistochemical validation of BMI-1 or any of the marker gene signatures on human tumor samples were presented.

We thank Reviewers #1-2 for having addressed this issue, which was also brought up by Reviewers #3 and #4. We have performed patients’ samples next-generation sequencing and immunohistochemistry validation for BMI-1, which we have included in the current version (please see **Supplementary Data, and Supplementary Figures 2l and 2n**).

Comments to the Authors:

This paper is reviewed in the context of the urgent need to have targeted therapy for KRAS mutant lung tumors with associated biomarkers to allow precision medicine. It is also reviewed in the context of many papers reporting scRNAseq data on relatively small numbers of patients (like this report) with non-small cell lung cancers and trying, like the current study, to use this information to both understand the pathogenesis of non small cell lung cancer, and to identify new targets for therapy particularly in KRAS mutant NSCLCs. While the study was technically well done, it was presented in a complex and unwieldy manner that made it hard to follow. The authors need to address several issues:

1. *They need to focus on the 10 lung adenocarcinomas since squamous cancers almost never have KRAS mutations. The two squamous patients do not add any thing to the analyses. In addition, they need to have their text and patient table correlate – they have 7 KRAS mutant and 3 wild type (not 2 as stated) LUADs. There were*

several other examples, where their complex writing style led to having to “track things down” which was not desirable to the reader.

We apologize for not making it clear that when analyzing clinical samples according to their KRAS mutation status we only used the 10 lung adenocarcinomas, therefore excluding the squamous samples, as Reviewers #1-2 also requested.

We also thank the Reviewers for having noticed that **Table S1** and its related text did not correlate, thus in the current version we included the correct number of KRAS wild type patients.

Finally, we took to heart their suggestions to improve the manuscript readability, thus we have restructured some sentences accordingly, and simplified existing figures (**Figure 2b and 3f**).

2. They need to provide simple demographics about smoking status of the 10 LUADs they studied.

This information has now been included in **Supplementary Data**.

3. They found 15 different clusters in humans and 13 clusters in mouse materials. We need to know in a simple manner how did each of the human clusters map to (or not map to) a mouse cluster. We are left thoroughly confused about this simple fact.

When attempting to computationally ascertain how many biologically relevant transcriptomic clusters of cells are found in our samples, a mathematical dimensionality reduction technique called “Principle Component Analysis” (PCA) was used. When using PCA, the mathematical model can report how many components/dimensions best describe the variability seen in our dataset. We use this number of significant components/dimensions as a surrogate value for the number of biological clusters that can be found in our dataset. For the datasets used in our study, the human dataset was found to contain 15 transcriptomic clusters, while the mouse dataset only showed 13 clusters. The numbers of transcriptomic clusters identified per species is not arbitrary but rather driven by the biological variation inherent in the dataset. As such, the human 15 clusters and the murine 13 ones better described the variability observed. These numbers also are consistent with knowledge of the underlying biology wherein human adenocarcinoma tumor samples would most certainly be more transcriptionally variable than clonally bred mice.

In addition, as the human and mouse data were processed independently of each other, the number of clusters in each sample do not translate across, ie cluster 1 in the human sample does not correspond to cluster 1 in the mouse samples. The designation as to which cluster gets which number is arbitrary. It was completely coincidental that the numbering as cluster 10 for both mouse and human datasets matched. This was not intentional but rather a curious coincidental result.

We also understand and share the Reviewers’ interest in relating each human and murine cluster to each other; however, the available human epithelial annotation is not as accurate as the murine one, and it has not improved since initial submission of the manuscript. Therefore, we are still unable to compare clusters, as we would have wished ourselves. While murine epithelial cells can be subdivided into Alveolar (Bipotent, Type I and Type II), basal, ciliated, and club cells, human samples can only be annotated as three separate epithelial clusters (C2, C5 and C10), whose function has not been annotated yet. The only parallelism we have been able to perform with a high degree of accuracy and confidence is indeed the one related to tumor Cluster 10, as already described in the manuscript.

4. It appears that human cluster 10 was the key tumor cluster and there was a mouse cluster that mapped to this. If there are other human clusters that contain tumor cells we need to know and did they map to any mouse clusters?

Also we know there can be intra-tumor heterogeneity (ITH) in LUAD – was there such ITH found here, and was this represented by Cluster 10 and some other tumor clusters? Again, this simple fact is not easily discernible.

Cluster 10 is the cluster that we were able to match among human and murine samples with greatest accuracy and confidence owing to the limitations of current human single-cell transcriptomic annotation atlas'. In Figure 1B we can appreciate in green all the tumor epithelial cells identified in patients' samples; they can be further subdivided into three different clusters: C2, C5 and C10. While C10 is detectable in KRAS mutant patients, C2 and C5 are present in all patient samples. Regrettably, as mentioned in our response above, the available human annotation is not as defined as the murine one; therefore, we have not been able to obtain a more defined characterization.

To address the intra-tumor heterogeneity (ITH) topic, this is certainly an excellent point, and we agree with the Reviewers that such heterogeneity is observed in our dataset. After a thorough analysis, following also the Reviewers' request to ascertain patient's tumor burden, we observed that clusters are differentially represented across patient samples. Clusters 2 and 5, despite being present in all patient samples, independently of their KRAS mutation status, are represented at varying percentages. We have included a statement in the Discussion addressing this matter, also emphasizing that despite the expected (and observed) human ITH, we have been able to identify Cluster 10 as enriched only in patients carrying KRAS activating mutations, as compared to KRAS wt patients (2.77% versus 0.13%; FDR 2.07×10^{-116} , **Supplementary Table 2**).

5. There is discussion of 21 up and 14 down regulated genes presumably in Cluster 10. But then we hear about 50 down regulated genes. Which is correct?

We thank the Reviewers #1-2 for requesting this clarification, which led us to correct an inaccuracy. There are indeed 21 upregulated genes ($\text{LogFC} \geq 1.5$) and 9 (not 14) downregulated genes ($\text{LogFC} \leq 1.5$). We apologize for that slip-up, and corrected both text and heat maps accordingly (**Figure 3a**). Fourteen genes are the number of common genes within the top 100 downregulated genes, a parameter we had previously adopted, which was however less stringent than considering the $\text{LogFC} \leq 1.5$. Thus, after correcting the heatmap with 9 downregulated genes, we were glad to realize that the signature did not change considerably (**Figure 3**), thus the main conclusions we had reached in the original manuscript were unaffected. **Figures 3d** and **3e** are meant to provide representative expressions of specific marker genes contained within the signature. In particular they represent genes comprised within the 50 highest or lowest genes, commonly present in both the human and murine dataset.

6. All of their preclinical studies were done with the old “war horse” for NSCLC studies A549 cells. Of course A549 is wild type for TP53 (while the GEMM models are TP53 and KRAS mutant) making them not comparable. In addition, A549 cells are STK11 mutant, which has major implications for tumor behavior. None of these aspects were controlled. Of course, the usual things would be to have several xenograft models, and potentially patient derived xenograft (PDX) models that were tested.

This a relevant question, which we addressed in the current manuscript. While it is true that in the original manuscript we presented only *in vivo* and in culture data obtained with the KRAS mutant and TP53 wt A549 cell line (also mutant for STK11), following the Reviewers' suggestions we assayed the KRAS mutant and TP53 mutant cell line SKLU1 (which is instead wt for STK11). With respect to their concern that SKLU1 may better model the murine system (being TP53 and KRAS mutant), we were able to show that a different TP53 mutation status did not impact their response to PTC596 treatment in culture, as compared to A549 cells. These data imply that drug response is TP53-independent. These data have been included in **Figures 4a-f**, and mentioned in the Discussion paragraph.

STK1 is not the focus of this manuscript, however, to address the Reviewers' request, our data also show that a

different STK1 status (mutant in A549 and wt in SKLU1 cells) does not appear to affect drug response. Xenograft assays could not be performed by adopting SKLU1 cells as this line is known for its inability to grow subcutaneous tumors (<https://www.mskcc.org/research-advantage/support/technology/tangible-material/sk-lu-1-human-lung-cell-line>). However, we here share with the Reviewers (also Reviewer #4 had a similar question) data obtained while working on a different project, supporting the fact that the KRAS-signature is TP53-status independent. Our data show that a pulmonary epithelial tumor-associated subpopulation carrying the KRAS-signature described in this manuscript, was also detectable in mice only carrying mutant KRAS (K mice) (**Figure for Reviewers 1a**). Overall, these data imply that the KRAS signature is TP53-independent and KRAS-associated. In addition, we are also showing the Reviewers that KRAS mutant mice treated with PTC596 for up to 4 weeks respond to PTC596 treatment, implying the drug response is unrelated to TP53 status (**Figure for Reviewers 1b-c**).

Unfortunately, we did not have PDX models available but we hope these data will strengthen the data that support the ability of PTC596 ability to affect KRAS mutant tumors.

7. The xenograft studies were done on very small tumors (they report starting treatment with size “2 to 3 mm” diameter tumors). While there is nothing “wrong” with this, typically one would start treatment on larger size tumors.

This is an appropriate comment, however our xenograft studies were actually started “once subcutaneous tumors reached ~80-90 mm³”. The “1-2 mm diameter” measurement was instead referred to tumors measured by MRI in transgenic mice. We have made sure this is clearly stated throughout the entire revised manuscript.

8. It is very hard to know how much of an anti-tumor effect the PTC596 treatment had. They use words like “virtual complete disappearance” of a tumor cell population by scRNAseq analyses. But looking at Figure 5 f. the % tumor volume goes from 100 to 80%. Also what does “tumor volume” on the y-axis of figure 5 a mean?

We apologize for not making it clear in our initial submission that lung tumor burden in mice was presented as percentages, with tumor volume set at 100% at day 1 of treatment (as shown on the y-axis in **Figure 5f**). To answer the Reviewers’ question on the anti-tumor effect of PTC596, we agree that **Figure 5f** shows MRI quantifications indicating that drug-treated tumors go from 100% to ~80% of their initial volume, thus showing an overall ~16% decrease in tumor size at 4 weeks of treatment. Following your suggestion, we have made this clearer in the revised manuscript.

However, we also observed that at 4 weeks of treatment the Vehicle-treated group showed a tumor increase to up to 156.1% (tumor volume was set at 100% at day 1 of treatment), while the Drug-treated tumors shrunk to 83.7% (1.87-fold reduction; FDR=2.36x10⁻⁷) (**Figure 5f**). Thus, we commented that such reduction in tumor size, as detected by MRI, was similarly accompanied by a ~2-fold decrease in the percentage of total tumor epithelial cells detected by scRNAseq. In fact, while 47.9% of Vehicle-treated tumor cells are epithelial cells, such percentage decreases to 22.2% (2.16-folds change; FDR=5.75 x10⁻⁵⁷) (**Figure 5c**) in PTC596-treated tumors.

We also made the observation that within the totality of tumor epithelial cells, C10 was, percent-wise, the most strongly affected, in PTC596-treated tumors. However, following your recommendation, we removed the statement “virtual complete disappearance” of C10, and substituted with “the murine C10 subpopulation was almost completely abrogated (5.4-fold reduction, from 11.8% to 2.2%, FDR= 3.96 x10⁻⁶⁷) in Drug- versus Vehicle-treated tumors” (**Figure 5e**).

9. It would have been important to know the mutation status of many genes they would have from their RNAseq data to find out about other key mutations (e.g. TP53, STK11 to name a few) and also the tumor mutation burden, and expression of PDI, PDL-1 on tumor cells and in the tumor microenvironment. The reasons for

these studies are obvious given both the widespread use of CLIA certified mutation testing on NSCLC, and the movement of checkpoint inhibitor therapies into first line treatment of NSCLCs.

Studies related to PD1 and PDL1 are certainly important in oncology research; however we could not identify any relevant gene expression information in our epithelial subpopulations, which represent the focus of these studies.

However, by following Reviewers #1-2 request, which was also shared by Reviewers #3 and 4, we obtained the mutation burden information of all the adenocarcinoma patients utilized in this study, which has been uploaded as **Supplementary Data**. Since the single cell RNAseq technique we utilized (InDrop) only generates reads at the 3' UTR [1], not allowing analysis of the overall mutation status, we extracted genomic DNA from our patients' sample and sequenced them. Such detailed analysis allowed us to confirm and improve the information we are now able to provide. In the previous version of the manuscript, we divided the patients in KRAS mutant and WT, based on the results of the ddPCR. Now, we have performed targeted next-generation sequencing to confirm potential mutations in KRAS. Our data show that both groups of patients (KRAS activating mutation carriers and KRAS wild-type) were correctly identified.

We were also able to include an extra patient in our analysis. Originally patient NSC40 paraffin-embedded tissue did not yield sufficient DNA to allow analysis of *KRAS* mutation status, therefore sample NSC40 was not utilized to generate **Figure 1f**. For the resubmission, we managed to obtain sufficient material for sequencing and also this patient (carrying KRAS activating mutations) confirms our observation that C10 is indeed present only in mutant samples.

10. Since BMI-1 is a key cancer stem cell expressed gene, it would have been interesting to know if PTC596 treatment actually removed the cancer initiating cells in either clonogenic assays, monitoring other cancer stem cell markers (such as ALDH related genes), or even better in serial transplantation studies. Also whether they did this equally well in KRAS mutant vs. KRAS wild type tumors. This type of information would be important for precision medicine clinical translation.

We agree this is a good point. PTC596 treatment does indeed significantly affect murine Cluster 10 from KP mice. Regrettably, unrelated studies performed on a different lung cancer model (driven by mutant EGFR, and thus wt for KRAS) do not contain a C10-like subpopulation, nor they carry any KRAS-associated signature (data not shown). However, in order to address the Reviewers' question whether C10 may contain cancer-initiating cells we performed RNA velocity analysis. Such analysis, in combination with single-cell trajectory study, can provide insights into the transcriptional dynamics of cells' evolution [2]. Interestingly, as shown in **Figure 2g**, C10 contains a point of origin (short/no arrows, indicated by the dotted box) that appears to give rise to the transformed epithelial cells, thus suggesting C10 might indeed contain tumor-initiating cells. In addition, gene expression profiling and gene set enrichment analysis (GSEA) of C10 cells versus all other tumor epithelial cells, showed enrichment of i) stem cell signatures (**Figure 2f**), ii) stemness genes (stem cell, embryonic, hematopoietic, mammary stem cells, liver cancer stem cells, cancer radiotherapy responsiveness) (**Supplementary Figure 1d**) as well as target genes of the cancer stem cell gene BMI1 (**Figure 2f and Supplementary Figure 1d**). These data corroborate the hypothesis, suggested by the Reviewers, that C10 might indeed contain tumor-initiating cells.

Reviewer 3

Comments:

1) This manuscript describes a subset of lung adenocarcinomas with a specific molecular signature. RNA sequencing was performed, and the effects of in vivo PTC596 treatment, which affects BMI-1 activity, was

evaluated in the murine model.

Lung cancer is a common disease with a poor prognosis. In adenocarcinomas of the lung, KRAS-mutations are typically found in 1/3 of the tumors. Up to now, there has been efforts in developing treatment specifically targeting KRAS, but no such treatments are implemented in clinical practice yet. However, a very promising drug have shown exquisite results for lung cancer patients with the G12C mutation treated with AMG 510. A mention of this would improve the introduction.

This information has now been included in Introduction.

2) The investigations performed molecular analyses including sequencing, cell lines studies and mouse model work. The analyses give interesting results and seem to be thoroughly done. Some analyses are performed on tumor tissue from lung cancer patients with KRAS-mutation in their tumor. It would be interesting to know which KRAS-mutation where found, and several papers have reported different biology depending on the different KRAS-mutations found. Were the samples also analyzed for other mutations? How was the tumors selected?

We thank Reviewer #3 for his/her appreciation of our work. Following his/her suggestion (as well as the Other Reviewers), we have performed genome sequencing to describe patients' tumor burden. This information has now been included as **Supplementary Data**. Upon our analysis, we did not identify biological differences depending on the different KRAS mutations identified, however our new next-generation sequences validate the original observation that only patients with KRAS activating mutations contain C10 cells carrying the KRAS-associated signature.

In addition, to answer the question from Reviewer #3 on how tumors were selected: we utilized in an unbiased manner all the clinical resected material that we have been able to obtain from the clinics. Patients were selected when having tumors greater than or equal 1.6 cm that were treatment naïve. This information has now been included in the Material and Method Section, per the Reviewer's request.

3) The authors call it a subpopulation, but the signature seems to be present in all the KRAS-tumors, in contrast to normal cells.

Please allow us to specify that the signature was identified by comparing C10 overall gene expression versus all other tumor epithelial cells', therefore the gene expression data we are showing are specifically present in C10. We have rewritten this paragraph, wishing to have made it clearer.

4) It would be very interesting to see the MRI images of the mice, - showing examples of the shrinking. The authors call it shrinking and at the same time 50% reduced tumor growth. Figure 5f shows a reduction in tumor volume. I would then suggest not to call it tumor growth, as one can get the impression that the growth rate is reduced, but it is still growing. Were any mutational analyses performed on the tumors after treatment?

Please find the images you request in the **Supplementary Figure 4a**.

In addition, to follow the Reviewer's suggestion we have rewritten the sentence and removed the word "growth". Now it reads "At day 28 of PTC596 treatment tumors are shrunk by 16.3%, as compared to day 1. In addition, while tumor volume increased up to 156.1% by day 28 in the Vehicle-treated group, drug-treated tumors shrunk to 83.7%, resulting in a significant 46.3% reduction in tumor size (FDR=2.36x10⁻⁷)".

Also, in relevance to any mutational analysis the Reviewer is referring to, given the RNA sequencing technique that we adopted (Indrop), which does not produce a full-length transcript [1] we were unable to follow up mutation status upon treatment. This is an excellent point, and future studies will adopt newer techniques that allow detecting mutations in all our samples.

5) How was the "take" in the mouse model work?

The criteria we adopted to enroll transgenic mice into either Vehicle or Drug treatment was selecting mice carrying tumors of similar size (1 to 2 mm diameter) by screening them by MRI before treatment initiation. This data is now included in the Material and Method section.

Reviewer #4

Remarks to the Authors:

This manuscript by Maroni et al reports the identification of a mutant K-RAS associated gene signature, which is conserved in both human and murine lung cancers. The authors show that PTC596, a BMI inhibitor, inhibits tumor growth in xenograft and mutant K-RAS/p53-null (KP) mouse models with concomitant elimination of C10 subpopulation of epithelial cells. As there is no targeted therapy approved for mutant K-RAS-mediated NSCLC, identification of a tumorigenic and targetable subpopulation may have profound impact in designing improved therapies.

This is an interesting work and the conclusions are broadly supported by the experimental data provided. The authors have done a good job in verifying new observations, pursuing orthogonal methods to confirm specific results, and drawing meaningful conclusions. Overall, I do believe that this work is of the caliber to be published in this journal. The data presented in this study are clear, well-presented, statistically significant, and adequate controls are included in the experiments.

We thank Reviewer #4 for his/her appreciation of our work.

Comments:

There are some unanswered questions and this study would be further strengthened by addressing the following:

1) The authors have made a key statement that there is a tumor enriched epithelial cluster, C10, which is specific for mutant-KRAS adenocarcinomas (ADCs) compared to WT K-RAS ADCs. In lung ADCs, mutations in K-RAS and p53 co-occurs in 40-45% cases, and p53 mutations takes place in almost double the number of K-RAS mutant cases. This C10 cluster is also present in tumors from combined mutant KRAS and p53-null mouse (KP). Therefore, is it possible that this cluster of epithelial cells is conserved for KRAS and p53 loss-of-function mutations? Similarly, the associated transcriptomic signature should be attributable to both K-RAS and p53 mutations in lung cancer. Along this line, it is better to mention how many of the seven K-RAS mutant patients have also p53 mutations (Supplementary Table 1).

This is certainly a good point, which was also raised by Reviewers #1-2. We addressed this question by analyzing our KRAS mutant patients. We observed that C10 is present in 67% of TP53 mutant patients and 80% of TP53 wt patients, thus suggesting C10 is not related to TP53 mutant status.

In addition, we are showing to Reviewer #4 data we generated on different murine models, showing the KRAS-signature is TP53-status independent and KRAS-specific. In fact, a subpopulation carrying the KRAS-signature observed in Cluster 10 from the KP mice, can also be detected in mice only carrying mutant KRAS (**Figure for Reviewers 1a**), whereas it is undetectable in EGFR mutant mice (data not shown). We hope these data will increase the Reviewer's trust that C10 is related to mutant KRAS status, and it is TP53 status independent.

2) Ingenuity Pathway Analysis (IPA) identified few signaling pathways like EIF2, mTOR and eIF4/p70S6K as the top three enriched pathways in human and mouse C10 populations. In addition, IPA detected activation of

other pathways including Integrin, EGF, IGF-1, MEK/ERK, and Insulin receptor in the C10 population from human and mouse K-RAS associated lung tumors. The rationale for choosing the oncogene BMI-1 as a target for treating this C10 population of tumor cells is not clear, although the mutant K-RAS ADC cells are positive for it, as suggested by the authors. BMI-1 is not listed in the top marker genes in tumor tissues (Supplementary Table 4) and it is not among the top 50 highest expressing genes in the transcriptional profile C10 (Supplementary Table 7). Therefore, why BMI-1 was chosen as the target for this K-RAS specific cluster, and not other activated pathways or genes that are on the top of the list.

This is an appropriate observation. The logic we followed was the following: We had previously investigated the relevance of BMI1 in lung cancers driven by low/negative C/EBP α expression [3]. Therefore, we asked whether BMI1 might also be relevant in KRAS mutant lung cancer. We were impressed by the BMI1 high positivity in KRAS mutant lung cancer patients that we observed by immunohistochemistry. When we started the murine studies by performing scRNASeq analysis we did not know *a priori* of the existence of C10, we just discovered it afterwards. We also discovered though that C10 is not the only subpopulation that is shrinking upon PTC596 treatment, although it is the one that gets shrunk to the lowest percentage. In terms of BMI1 expression, its transcript is not expressed among the top 50 highest expressing genes, however when performing a GSEA between C10 against all other tumor epithelial clusters, C10 shows enrichment of BMI1 target genes, implying BMI1 activity is relevant in C10.

Of course, we agree that future studies should also be aimed at studying the effects of inhibiting other pathways active in C10 (and other tumor epithelial subpopulations).

3) It is important to discuss whether BMI-1 is a downstream target of any of those signalings mentioned above, which are activated in the K-RAS specific C10 epithelial population. It is worth mentioning for the differential expression of all of the BMI-1 upstream regulators (like MYC, SALL4, Nrf2, etc) or downstream targets (like Ink4a, Arf, PTEN, Hox, VEGF, etc) in the scRNA-seq data from C10 population with or without the treatment by PTC596. As BMI-1 is a regulator of cancer cell stem phenotype, what is the status of expression of stemness/differentiation markers in this cell population before and after treatment?

This is certainly a relevant question, however we could not technically observe any significant difference in gene expression between Vehicle- and PTC596-treated cells, as C10 is decreased to few cell (n=69), thus not allowing to perform a statistically relevant analysis. However, we thank the Reviewer #4 for having asked the question related to the expression of stemness genes in C10. We were able to address it and interestingly, gene expression profiling of C10 cells versus all other epithelial cells and gene set enrichment analysis (GSEA) showed enrichment of adult stem cell signatures (**Figure 2g**), of stemness genes (stem cell, embryonic, hematopoietic, mammary stem cells, liver cancer stem cells, and cancer radiotherapy responsiveness) (**Supplementary Figure 1d**), as well as enrichment of the cancer stem cell gene BMI1 targets (**Figure 2g and Supplementary Figure 1d**).

4) BMI-1 activates downstream oncogenic functions through its upregulation in the tumor. But here, the authors did not observe appreciable change in its expression in the C10 population, rather other oncogenic pathways like MEK/ERK, mTOR, NF- κ B, and p70S6K are strongly activated. Therefore, the C10 population of tumor cells can be best targeted by MEK/ERK, mTOR, and p70S6K inhibitors that are already approved. To clarify which inhibitor targets these cells more efficiently, it is better to compare the efficacy of PTC596 with that of MEK/ERK and p70S6K inhibitors in xenograft assays with lung tumor cell lines. In addition, tumor shrinkage with PTC596 is only 50% in the KP mouse model. It will be interesting to determine if the combination of PTC596 with MEK/ERK, mTOR, NF- κ B or p70S6K inhibitor can reduce the tumor growth completely with concomitant abolishment of murine C10 subpopulation. These experiments can be done using xenograft models

This is an excellent point. However, since C10 has been identified in transgenic mice, in future studies we will test the inhibitors mentioned by Reviewer #4 as single agents or in combinations, in KP mice. Since we also considered that MEK inhibition might be relevant in KP tumors growth, we prioritized testing its inhibition by adopting Selumetinib (an allosteric MEK1/2 inhibitor) as both single agent and in combination with PTC596. Remarkably, our MRI data show that PTC596 alone achieves better results than Selumetinib alone, and that their combined inhibition is not synergic, in that adding Selumetinib does not significantly improve the effects achieved by PTC596 alone (**Supplementary Figure 4b**). These data therefore suggest that BMI1 inhibition alone is better than MEK inhibition, and different pathways will have to be further tested (as also suggested by Reviewer #4) to identify more efficient combinations.

5) Although BMI-1 is not upregulated in the human and mouse tumor enriched cluster (TEC) C10, its inhibitor has significant effect on the tumor regression with corresponding reduction of this population. It raises an important question whether PTC596 inhibits any of the activated pathways like mTOR, p70S6K, MEK/ERK, EGF, NF-kB, etc. in the C10 population. These can be shown in lung cancer cell culture studies.

This is a good question; however, we cannot utilize cell lines to study C10 cells, as they were identified in tumors growing *in vivo*. Similarly, it is complicated to study which pathways are inhibited by PTC596 in drug-treated C10, as this cluster significantly shrinks and the few remaining cells (n=69) are not sufficient to perform a statistically appropriate pathway analysis. However, since we took at heart Reviewer #4's suggestion, we checked by Ingenuity Pathway Analysis the only dataset available to us, that could technically generate statistically relevant observations on PTC596-treated tumor cells. In particular, by adopting scRNAseq data generated on adenocarcinomas from EGFR mutant mice (their tumor epithelial cells are Bmi1⁺), our analysis show that p70S6K, mTOR, ERK/MAPK, and EGF signaling, are indeed predicted to be inhibited upon PTC596 treatment (**Figure for Reviewers 1d**).

6) It is better to provide the mutational status of important genes in patient lung tumors in the Supplementary Table 1, if available.

Following Reviewer #4's and the Other Reviewers' suggestion we have sequenced all the clinical specimens and included the entire mutation status as **Supplementary Data**. Importantly our next-generation sequencing data confirmed our previous data, obtained by ddPCR, on the KRAS mutation status thus not altering our original observation.

7) A brief description of the BMI-1 protein and its function in the 'Introduction' of the manuscript will be helpful.

We have included new information on BMI-1 in the Introduction.

8) In some figures labeling is very small and illegible. It should be fixed.

We thank Reviewer #4 for this suggestion and we fixed it by enlarging them all.

REFERENCES

1. Klein, A.M., et al., *Droplet barcoding for single-cell transcriptomics applied to embryonic stem cells*. Cell, 2015. **161**(5): p. 1187-201.
2. La Manno, G., et al., *RNA velocity of single cells*. Nature, 2018. **560**(7719): p. 494-498.
3. Yong, K.J., et al., *Targeted BMI1 inhibition impairs tumor growth in lung adenocarcinomas with low CEBPalpha expression*. Sci Transl Med, 2016. **8**(350): p. 350ra104.

Figures for Reviewers

Kras mice

1c

1d

Figure 1 for reviewers. **a)** SPRING plots showing the positive (**left panel**) and negative (**right panel**) component of the mutant-KRAS-associated signature in KRAS mutant pulmonary tumors, that are wt for TP53 status. For the positive/upregulated (**left panel**) and the negative/downregulated signature (**right panel**) the more genes detected per cell, the stronger the enrichment score, represented as a scale from 0 (grey) to 0.5 (blue) to 1 (red), where an enrichment score of 1 signifies detected expression of all marker genes within that cell. **b)** Representative pictures of KRAS mutant (TP53 wt) tumors (indicated by the arrow), as imaged by MRI in a Vehicle- and a PTC596-treated mouse, at day 1 and 28 of treatment. **c)** MRI quantification of Kras mice treated for 28 days with Vehicle (grey line) and PTC596 (black line). **d)** IPA analysis showing that p70S6K, mTOR, ERK/MAPK, and EGF signaling are predicted to be inhibited upon PTC596 treatment in EGFR mutant (Bmi1⁺) pulmonary tumors.

Sincerely,

Elena Levantini, Ph.D.
Instructor of Medicine
Harvard Medical School
Research Associate
Hematology/Oncology
Beth Israel Deaconess Medical Center
Life Science Center, 3 Blackfan Circle 428-F12
Boston, MA 02215
Tel 617 735 2214
Fax 617 735 2222
Email: elevanti@bidmc.harvard.edu

Reviewers' comments:

Reviewer #1 (Remarks to the Author):

The authors have responded appropriately to the reviewers' comments including providing substantial additional experimental data.

Reviewer #2 (Remarks to the Author):

In this revised manuscript, the authors have addressed most of our comments. The additional results are interesting and the clinical annotation is helpful. However, we hope the authors could still consider the following comments for further improvement of the manuscript.

1. Previously we suggested the authors focus on the LUAD samples only. The authors clarified that analyses according to KRAS mutation status were performed for LUAD samples only. However, it is to my understanding that the clustering analyses were still performed with all 12 samples, including the two LUSC samples. This may confound the comparison with the mouse KP ADC tumors. If the authors wish to do so, I think it will be helpful to generate 12 SPRING plot each with dots colored to highlight cells from one patient and indicate the histology (LUSC or LUAD) and KRAS mutation status in the legend. They can also provide a frequency table (in %) with cluster or cell type in rows and patient ID in columns.

2. We asked the authors to provide some kind of matching for the human and mouse clusters. The authors explained the number of clusters are different as they were determined in a data-driven way, which is understandable. They may consider using the mouse orthologues of human cluster signature to cluster the mouse cells and see how the resulting cell clusters match with the mouse signature based cluster, and do the opposite for human cells. Results can be presented as heatmaps and/or crosstabs.

3. The authors were also unable to find accurate human epithelial annotation. There is a resource that just got published (<https://www.nature.com/articles/s41586-020-2922-4>). The authors may consider using the markers identified in this paper.

4. Supplementary Figure 1d, the authors should report normalized enrichment score(NES) and FDR q value like they did for Supplementary Figure 2e-g.

5. Providing merged and normalized data (instead of one file per sample) and cell cluster assignment annotation in the GEO repo will help with data reuse.

Reviewer #3 (Remarks to the Author):

The authors have adjusted the manuscript according to feed-back.

Reviewer #4 (Remarks to the Author):

In this revised manuscript, the authors satisfactorily addressed all my questions. I feel that the revised manuscript has been improved much in response to reviewers' critiques, and it is now suitable for publication in Communications Biology. The statistical analyses are appropriate and the experimental details are sufficient to reproduce the data.

ELENA LEVANTINI, Ph.D.

Center for Life Sciences

3 Blackfan Circle
Boston, MA. 02215

Tel: 617-735-2233 • **Fax:** 617-735-2222

E-mail: elevanti@bidmc.harvard.edu

January 15th, 2021

We appreciate that already three of the four original reviewers recognized the strength of our manuscript by endorsing its publication. We are also appreciative they noticed the substantial additional experimental data we have included, indicating that both the statistical analyses and the experimental details are appropriate to ensure reproducible data. We are therefore pleased about the possibility of publishing our study after we also address the additional suggestions provided by Reviewer #2, thus further strengthening our manuscript, as indicated below:

Reviewer #2

Remarks to the Authors:

In this revised manuscript, the authors have addressed most of our comments. The additional results are interesting and the clinical annotation is helpful. However, we hope the authors could still consider the following comments for further improvement of the manuscript.

We thank Reviewer #2 for his/her appreciation of our work. Following his/her suggestion, we have performed the requested analyses and we are glad to report they confirm the original observations described in our manuscript (please see below for details).

Comments to the Authors:

1. Previously we suggested the authors focus on the LUAD samples only. The authors clarified that analyses according to KRAS mutation status were performed for LUAD samples only. However, it is to my understanding that the clustering analyses were still performed with all 12 samples, including the two LUSC samples. This may confound the comparison with the mouse KP ADC tumors. If the authors wish to do so, I think it will be helpful to generate 12 SPRING plot each with dots colored to highlight cells from one patient and indicate the histology (LUSC or LUAD) and KRAS mutation status in the legend. They can also provide a frequency table (in %) with cluster or cell type in rows and patient ID in columns.

Taking into account the Reviewer's indication, we have analyzed and clustered only the LUAD clinical samples. As shown in **Figure for Reviewer 1a** we were able to identify a cluster which is enriched only in patients carrying KRAS activating mutations (C3), as compared to KRAS wt patients (2.77% versus 0.06%; $p = 7.52 \times 10^{-90}$). This cluster is also positive for the KRAS-associated signature identified in this study, as expected, thus not altering our original observation (**Figure for Reviewer 1b**).

We have also included a new PCA-reduced visualization into the main text's supplemental, in which each sample is highlighted by a distinct color, to better visualize each patient's contribution, per the reviewer's request (**Supplementary Figure 1**).

2. *We asked the authors to provide some kind of matching for the human and mouse clusters. The authors explained the number of clusters are different as they were determined in a data-driven way, which is understandable. They may consider using the mouse orthologues of human cluster signature to cluster the mouse cells and see how the resulting cell clusters match with the mouse signature based cluster, and do the opposite for human cells. Results can be presented as heatmaps and/or crosstabs.*

We thank the Reviewer for agreeing on our data-driven approach. In order to comply with his/her request to better clarify what each murine and/or human cluster contains, we have included a new Supplementary Table (**Supplementary Table 9**) in which predominant cell populations are shown per each murine and/or human cluster as well as cell types across both species.

3. *The authors were also unable to find accurate human epithelial annotation. There is a resource that just got published (<https://www.nature.com/articles/s41586-020-2922-4>). The authors may consider using the markers identified in this paper.*

We thank the reviewer for recommending this incredible new resource that has been recently published to the wider community. It will undoubtedly further our understanding and improve future investigations with such a detailed annotation atlas. Despite this however, as this paper was just published, it will be some time till its data is incorporated into the annotation packages we used for our analyses. We can write to the package developers to include it in future releases, however, the repository (Bioconductor) on which our annotation package is published, has a bi-yearly release cycle (Bioconductor - Release Announcements). Thus, it will be between 6-12 months till this annotation can be used to annotate datasets. As such, while it would be beneficial to include such an annotation, the findings within our work will not be altered by utilizing this atlas to annotate our human sample data and waiting for its incorporation into the annotation package used would delay our publication for 6-12 months with no significant gains or alterations to our manuscript.

4. *Supplementary Figure 1d, the authors should report normalized enrichment score (NES) and FDR q value like they did for Supplementary Figure 2e-g.*

We have revised accordingly as per the Reviewer's request.

5. *Providing merged and normalized data (instead of one file per sample) and cell cluster assignment annotation in the GEO repo will help with data reuse.*

We have updated the GEO uploads as requested by the Reviewer.

Figures for Reviewers

1a

KRAS-mutant Patients

KRAS-wt Patients

1b

KRAS-mutant Patients

Positive Signature

Negative Signature

Legends Figure 1 for reviewers.

a) SPRING plots displaying LUAD clinical samples. The red dotted box highlights cluster C3, enriched in KRAS-mutant patients (left panel) as compared to the KRAS-wt patients (right panel).

b) SPRING plots showing the common signature enrichment score for C3 cluster, calculated for each cell equivalent to the number of detected genes from the common signature. For the positive/upregulated and the negative/downregulated signature the more genes detected per cell, the stronger the enrichment score, represented as a scale from 0 (grey) to 0.5 (blue) to 1 (red), where an enrichment score of 1 signifies detected expression of all marker genes within that cell.

Sincerely,

Elena Levantini, Ph.D.
Instructor of Medicine
Harvard Medical School
Research Associate
Hematology/Oncology
Beth Israel Deaconess Medical Center
Life Science Center, 3 Blackfan Circle 428-F12
Boston, MA 02215
Tel 617 735 2214
Fax 617 735 2222
Email: elevanti@bidmc.harvard.edu

REVIEWERS' COMMENTS:

Reviewer #2 (Remarks to the Author):

The authors have satisfactorily addressed my questions in this version.